# GROUP EQUIVARIANT
# NEURAL POSTERIOR ESTIMATION

**Maximilian Dax**
Max Planck Institute for Intelligent Systems
Tübingen, Germany
`maximilian.dax@tuebingen.mpg.de`

**Stephen R. Green**
Max Planck Institute for Gravitational Physics
Potsdam, Germany
`stephen.green@aei.mpg.de`

**Jonathan Gair**
Max Planck Institute for Gravitational Physics
Potsdam, Germany

**Michael Deistler**
Machine Learning in Science, University of Tübingen
Tübingen, Germany

**Bernhard Schölkopf**
Max Planck Institute for Intelligent Systems
Tübingen, Germany

**Jakob H. Macke**
Max Planck Institute for Intelligent Systems &
Machine Learning in Science, University of Tübingen
Tübingen, Germany

## ABSTRACT

Simulation-based inference with conditional neural density estimators is a powerful approach to solving inverse problems in science. However, these methods typically treat the underlying forward model as a black box, with no way to exploit geometric properties such as equivariances. Equivariances are common in scientific models, however integrating them directly into expressive inference networks (such as normalizing flows) is not straightforward. We here describe an alternative method to incorporate equivariances under joint transformations of parameters and data. Our method—called group equivariant neural posterior estimation (GNPE)—is based on self-consistently standardizing the "pose" of the data while estimating the posterior over parameters. It is architecture-independent, and applies both to exact and approximate equivariances. As a real-world application, we use GNPE for amortized inference of astrophysical binary black hole systems from gravitational-wave observations. We show that GNPE achieves state-of-the-art accuracy while reducing inference times by three orders of magnitude.

## 1 INTRODUCTION

Bayesian inference provides a means of characterizing a system by comparing models against data. Given a forward model or likelihood $p(x|\theta)$ for data $x$ described by parameters $\theta$, and a prior $p(\theta)$, the Bayesian posterior is proportional to the product, $p(\theta|x) \propto p(x|\theta)p(\theta)$. Sampling techniques such as Markov Chain Monte Carlo (MCMC) can be used to build up a posterior distribution provided the likelihood and prior can be evaluated.

For models with intractable or expensive likelihoods (as often arise in scientific applications) simulation-based (or likelihood-free) inference methods offer a powerful alternative (Cranmer et al., 2020). In particular, neural posterior estimation (NPE) (Papamakarios & Murray, 2016) uses expressive conditional density estimators such as normalizing flows (Rezende & Mohamed, 2015; Papamakarios et al., 2021) to build surrogates for the posterior. These are trained using model simulations $x \sim p(x|\theta)$, and allow for rapid sampling for any $x \sim p(x)$, thereby amortizing training costs across future observations. NPE and other density-estimation methods for simulation-based inference (Gutmann & Corander, 2016; Papamakarios et al., 2019; Hermans et al., 2020) have been reported to be more simulation-efficient (Lueckmann et al., 2021) than classical likelihood-free methods such as Approximate Bayesian Computation (Sisson et al., 2018).

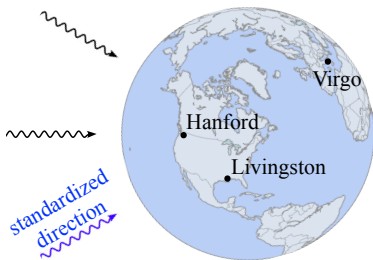

Figure 1: By standardizing the source sky position, a GW signal can be made to arrive at the same time in all three LIGO/Virgo detectors. However, since this also changes the projection of the signal onto the detectors, it defines only an *approximate* equivariance. Nevertheless, our proposed GNPE algorithm simplifies inference by simultaneously inferring *and* standardizing the incident direction.

Training an inference network for any $x \sim p(x)$ can nevertheless present challenges due to the large number of training samples and powerful networks required. The present study is motivated by the problem of gravitational-wave (GW) data analysis. Here the task is to infer properties of astrophysical black-hole mergers based on GW signals observed at the LIGO and Virgo observatories on Earth. Due to the complexity of signal models, it has previously not been possible to train networks to estimate posteriors to the same accuracy as conventional likelihood-based methods (Veitch et al., 2015; Ashton et al., 2019). The GW posterior, however, is equivariant[1] under an overall change in the time of arrival of the data. It is also *approximately* equivariant under a joint change in the sky position and (by triangulation) individual shifts in the arrival times in each detector (Fig. 1). If we could constrain these parameters *a priori*, we could therefore apply time shifts to align the detector data and simplify the inference task for the remaining parameters.

More generally, we consider forward models with known equivariances under group transformations applied jointly to data and parameters. Our aim is to exploit this knowledge to *standardize the pose of the data*[2] and simplify analysis. The obvious roadblock is that the pose is contained in the set of parameters $\theta$ and is therefore unknown prior to inference. Here we describe *group equivariant* neural posterior estimation (GNPE), a method to self-consistently infer parameters *and* standardize the pose. The basic approach is to introduce a *proxy* for the pose—a blurred version—on which one conditions the posterior. The pose of the data is then transformed based on the proxy, placing it in a band about the standard value, and resulting in an easier inference task. Finally, the joint posterior over $\theta$ and the pose proxy can be sampled at inference time using Gibbs sampling.

The standard method to incorporating equivariances is to integrate them directly into network architectures, e.g., to use convolutional networks for translational equivariances. Although these approaches can be highly effective, they impose design constraints on network architectures. For GWs, for example, we use specialized embedding networks to extract signal waveforms from frequency-domain data, as well as expressive normalizing flows to estimate the posterior—neither of which is straightforward to make explicitly equivariant. We also have complex equivariance connections between subsets of parameters and data, including approximate equivariances. The GNPE algorithm is extremely general: it is architecture-independent, it applies whether equivariances are exact or approximate, and it allows for arbitrary equivariance relations between parameters and data.

We discuss related work in Sec. 2 and describe the GNPE algorithm in Sec. 3. In Sec. 4 we apply GNPE to a toy example with exact translational equivariance, showing comparable simulation efficiency to NPE combined with a convolutional network. In Sec. 5 we show that standard NPE does not achieve adequate accuracy for GW parameter inference, even with an essentially unlimited number of simulations. In contrast, GNPE achieves highly accurate posteriors at a computational cost three orders of magnitude lower than bespoke MCMC approaches (Veitch et al., 2015). The present paper describes the GNPE method which we developed for GW analysis (Dax et al., 2021), and extends it to general equivariance transformations which makes it applicable to a wide range of problems. A detailed description of GW results is presented in Dax et al. (2021).

## 2 RELATED WORK

The most common way of integrating equivariances into machine learning algorithms is to use equivariant network architectures (Krizhevsky et al., 2012; Cohen & Welling, 2016). This can be in

---

[1]In physics, the term "covariant" is frequently used instead of "equivariant".

[2]We adopt the language from computer vision by Jaderberg et al. (2015).

conflict with design considerations such as data representation and flexibility of the architecture, and imposes constraints such as locality. GNPE achieves complete separation of equivariances from these considerations, requiring only the ability to efficiently transform the pose.

Normalizing flows are particularly well suited to NPE, and there has been significant progress in constructing equivariant flows (Boyda et al., 2021). However, these studies consider joint transformations of parameters of the base space and sample space—*not* joint transformation of data and parameters for *conditional* flows, as we consider here.

GNPE enables end-to-end equivariances from data to parameters. Consider, by contrast, a conditional normalizing flow with a convolutional embedding network: the equivariance persists through the embedding network but is broken by the flow. Although this may improve learning, it does not enforce an end-to-end equivariance. This contrasts with an *invariance*, for which the above would be sufficient. Finally, GNPE can also be applied if the equivariance is only *approximate*.

Several other approaches integrate domain knowledge of the forward model (Baydin et al., 2019; Brehmer et al., 2020) by considering a "gray-box" setting. GNPE allows us to incorporate high-level domain knowledge about approximate equivariances of forward models without requiring access to its implementation or internal states of the simulator. Rather, it can be applied to "black-box" code.

An alternative approach to incorporate geometrical knowledge into classical likelihood-free inference algorithms (e.g., Approximate Bayesian Computation, see (Sisson et al., 2018)) is by constructing (Fearnhead & Prangle, 2012) or learning (Jiang et al., 2017; Chen et al., 2021) equivariant summary statistics $s(x)$, which are used as input to the inference algorithm instead of the raw data $x$. However, designing equivariant summary statistics (rather than invariant ones) can be challenging, and furthermore inference will be biased if the equivariance only holds approximately.

Past studies using machine-learning techniques for amortized GW parameter inference (Gabbard et al., 2019; Chua & Vallisneri, 2020; Green & Gair, 2021; Delaunoy et al., 2020) all consider simplified problems (e.g., only a subset of parameters, a simplified posterior, or a limited treatment of detector noise). In contrast, the GNPE-based study in Dax et al. (2021) is the only one to treat the full amortized parameter inference problem with accuracy matching standard methods.

## 3 METHODS

### 3.1 NEURAL POSTERIOR ESTIMATION

NPE (Papamakarios & Murray, 2016; Greenberg et al., 2019) is a simulation-based inference method that directly targets the posterior. Given a dataset of prior parameter samples $\theta^{(i)} \sim p(\theta)$ and corresponding model simulations $x^{(i)} \sim p(x|\theta^{(i)})$, it trains a neural density estimator $q(\theta|x)$ to estimate $p(\theta|x)$. This is achieved by minimizing the loss

$$\mathcal{L}_{\text{NPE}} = \mathbb{E}_{p(\theta)} \mathbb{E}_{p(x|\theta)} \left[ -\log q(\theta|x) \right] \tag{1}$$

across the dataset of $(\theta^{(i)}, x^{(i)})$ pairs. This maximum likelihood objective leads to recovery of $p(\theta|x)$ if $q(\theta|x)$ is sufficiently flexible. Normalizing flows (Rezende & Mohamed, 2015; Durkan et al., 2019) are a particularly expressive class of conditional density estimators commonly used for NPE.

NPE amortizes inference: once $q(\theta|x)$ is trained, inference is very fast for any observed data $x_{\text{o}}$, so training costs are shared across observations. The approach is also extremely flexible, as it treats the forward model as a black box, relying only on prior samples and model simulations. In many situations, however, these data have known structure that one wants to exploit to improve learning.

### 3.2 EQUIVARIANCES UNDER TRANSFORMATION GROUPS

In this work we describe a generic method to incorporate equivariances under joint transformations of $\theta$ and $x$ into NPE. A typical example arises when inferring the position of an object from image data. In this case, if we spatially translate an image $x$ by some offset $\vec{d}$—effected by *relabeling the pixels*—then the inferred position $\theta$ should also transform by $\vec{d}$—by *addition* to the position coordinates $\theta$. Translations are composable and invertible, and there exists a trivial identity translation, so the set of translations has a natural group structure. Our method works for any continuous transformation group, including rotations, dilations, etc., and in this section we keep the discussion general.

For a transformation group $G$, we denote the action of $g \in G$ on parameters and data as

$$\theta \to g\theta, \tag{2}$$
$$x \to T_g x. \tag{3}$$

Here, $T_g$ refers to the group representation under which the data transform (e.g., for image translations, the pixel relabeling). We adopt the natural convention that $G$ is defined by its action on $\theta$, so we do not introduce an explicit representation on parameters. The posterior distribution $p(\theta|x)$ is said to be *equivariant* under $G$ if, when the parameter and data spaces are jointly $G$-transformed, the posterior is unchanged, i.e.,

$$p(\theta|x) = p(g\theta|T_g x)|\det J_g|, \qquad \forall g \in G. \tag{4}$$

The right-hand side comes from the change-of-variables rule. For translations the Jacobian $J_g$ has unit determinant, but we include it for generality. For NPE, we are concerned with equivariant posteriors, however it is often more natural to think of equivariant forward models (or likelihoods). An equivariant likelihood and an *invariant* prior together yield an equivariant posterior (App. A.1).

Our goal is to use equivariances to simplify the data—to $G$-transform $x$ such that $\theta$ is taken to a fiducial value. For the image example, this could mean translating the object of interest to the center. In general, $\theta$ can also include parameters unchanged under $G$ (e.g., the color of the object), so we denote the corresponding standardized parameters by $\theta_0$. These are related to $\theta$ by a group transformation denoted $g^\theta$, such that $g^\theta \theta_0 = \theta$. We refer to $g^\theta$ as the "pose" of $\theta$, and standardizing the pose means to take it to the group identity element $e \in G$. Applying $T_{(g^\theta)^{-1}}$ to the data space effectively reduces its dimensionality, making it easier to interpret for a neural network.

Although the preceding discussion applies to equivariances that hold exactly, our method in fact generalizes to *approximate* equivariances. We say that a posterior is approximately equivariant under $G$ if (4) does *not* hold, but standardizing the pose nevertheless reduces the effective dimensionality of the dataset. An approximately equivariant posterior can arise if an exact equivariance of the forward model is broken by a non-invariant prior, or if the forward model is itself non-equivariant.

### 3.3 GROUP EQUIVARIANT NEURAL POSTERIOR ESTIMATION

We are now presented with the basic problem that we resolve in this work: how to simultaneously infer the pose of a signal and use that inferred pose to standardize (or align) the data so as to simplify the analysis. This is a circular problem because one cannot standardize the pose (contained in model parameters $\theta$) without first inferring the pose from the data; and conversely one cannot easily infer the pose without first simplifying the data by standardizing the pose.

Our resolution is to start with a rough estimate of the pose, and iteratively (1) transform the data based on a pose estimate, and (2) estimate the pose based on the transformed data. To do so, we expand the parameter space to include *approximate* pose parameters $\hat{g} \in G$. These "pose proxies" are defined using a kernel to blur the true pose, i.e., $\hat{g} = g^\theta \epsilon$ for $\epsilon \sim \kappa(\epsilon)$; then $p(\hat{g}|\theta) = \kappa\left((g^\theta)^{-1}\hat{g}\right)$. The kernel $\kappa(\epsilon)$ is a distribution over group elements, which should be chosen to be concentrated around $e$; we furthermore choose it to be symmetric. Natural choices for $\kappa(\epsilon)$ include Gaussian and uniform distributions. For translations, the pose proxy is simply the true position with additive noise.

Consider now the posterior distribution $p(\theta, \hat{g}|x)$ over the expanded parameter space. Our iterative algorithm comes from Gibbs sampling this distribution (Roberts & Smith, 1994) (Fig. 2), i.e., alternately sampling $\theta$ and $\hat{g}$, conditional on the other parameter and $x$,

$$\theta \sim p(\theta|x, \hat{g}), \tag{5}$$
$$\hat{g} \sim p(\hat{g}|x, \theta). \tag{6}$$

The second step just amounts to blurring the pose, since $p(\hat{g}|x, \theta) = p(\hat{g}|\theta) = \kappa\left((g^\theta)^{-1}\hat{g}\right)$. The key first step uses a neural density estimator $q$ that is trained taking advantage of a standardized pose.

For an **equivariant** posterior, the distribution (5) can be rewritten as (App. A.2)

$$p(\theta|x, \hat{g}) = p\left(\hat{g}^{-1}\theta|T_{\hat{g}^{-1}}x, \hat{g}^{-1}\hat{g}\right)\left|\det J_{\hat{g}}^{-1}\right| \equiv p(\theta'|x')\left|\det J_{\hat{g}}^{-1}\right|. \tag{7}$$

For the last equality we defined $\theta' \equiv \hat{g}^{-1}\theta$ and $x' \equiv T_{\hat{g}^{-1}}x$, and we dropped the constant argument $\hat{g}^{-1}\hat{g} = e$. This expresses $p(\theta|x, \hat{g})$ in terms of the $\hat{g}$-standardized data $x'$—which is much easier to

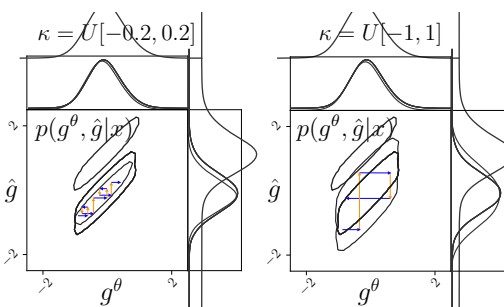

Figure 2: We infer $p(g^\theta, \hat{g}|x)$ with Gibbs sampling by alternately sampling (1) $g^\theta \sim p(g^\theta|x, \hat{g})$ (blue) and (2) $\hat{g} \sim p(\hat{g}|x, g^\theta)$ (orange). For (1) we use a density estimator $q(g^\theta|T_{\hat{g}^{-1}}(x), \hat{g})$, for (2) the definition $\hat{g} = g^\theta + \epsilon$, $\epsilon \sim \kappa(\epsilon)$. Pose standardization with $T_{\hat{g}^{-1}}$ is only allowed due to conditioning on $\hat{g}$. Increasing the width of $\kappa$ accelerates convergence (due to larger steps in parameter space), at the cost of $\hat{g}$ being a worse approximation for $g^\theta$, and therefore pose alignment being less effective.

estimate. We train a neural density estimator $q(\theta'|x')$ to approximate this, by minimizing the loss,

$$\mathcal{L}_{\text{GNPE}} = \mathbb{E}_{p(\theta)}\mathbb{E}_{p(x|\theta)}\mathbb{E}_{p(\hat{g}|\theta)}\left[-\log q\left(\hat{g}^{-1}\theta|T_{\hat{g}^{-1}}x\right)\right]. \tag{8}$$

With a trained $q(\theta'|x')$,

$$\theta \sim p(\theta|x, \hat{g}) \qquad \Longleftrightarrow \qquad \theta = \hat{g}\theta', \quad \theta' \sim q(\theta'|T_{\hat{g}^{-1}}x). \tag{9}$$

The estimated posterior is equivariant by construction (App. A.3).

For an **approximately-equivariant** posterior, (5) cannot be transformed to be independent of $\hat{g}$. We are nevertheless able to use the conditioning on $\hat{g}$ to approximately align $x$. We therefore train a neural density estimator $q(\theta|x', \hat{g})$, by minimizing the loss

$$\mathcal{L}_{\text{GNPE}} = \mathbb{E}_{p(\theta)}\mathbb{E}_{p(x|\theta)}\mathbb{E}_{p(\hat{g}|\theta)}\left[-\log q\left(\theta|T_{\hat{g}^{-1}}x, \hat{g}\right)\right]. \tag{10}$$

In general, one may have a combination of exact and approximate equivariances (see, e.g., Sec. 5).

### 3.4 GIBBS CONVERGENCE

The Gibbs-sampling procedure constructs a Markov chain with equilibrium distribution $p(\theta, \hat{g}|x)$. For convergence, the chain must be transient, aperiodic and irreducible (Roberts & Smith, 1994; Gelman et al., 2013). For sensible choices of $\kappa(\epsilon)$ the chain is transient and aperiodic by construction. Further, irreducibility means that the entire posterior can be reached starting from any point, which should be possible even for disconnected posteriors provided the kernel is sufficiently broad. In general, burn-in truncation and thinning of the chain is required to ensure (approximately) independent samples. By marginalizing over $\hat{g}$ (i.e., ignoring it) we obtain samples from the posterior $p(\theta|x)$, as desired.[3]

Convergence of the chain also informs our choice of $\kappa(\epsilon)$. For wide $\kappa(\epsilon)$, only a few Gibbs iterations are needed to traverse the joint posterior $p(\theta, \hat{g}|x)$, whereas for narrow $\kappa(\epsilon)$ many steps are required (Fig. 2). In the limiting case of $\kappa(\epsilon)$ a delta distribution (i.e., no blurring) the chain does not deviate from its initial position and therefore fails to converge.[4] Conversely, a narrower $\kappa(\epsilon)$ better constrains the pose, which improves the accuracy of the density estimator. The width of $\kappa$ should be chosen based on this practical trade-off between speed and accuracy; the standard deviation of a typical pose posterior is usually a good starting point.

In practice, we obtain $N$ samples in parallel by constructing an ensemble of $N$ Markov chains. We initialize these using samples from a second neural density estimator $q_{\text{init}}(g^\theta|x)$, trained using standard NPE. Gibbs sampling yields a sequence of sample sets $\{\theta_j^{(i)}\}_{i=1}^N$, $j = 0, 1, 2, \ldots$, each of which represents a distribution $Q_j(\theta|x)$ over parameters. Assuming a perfectly trained network, one iteration applied to sample set $j$ yields an updated distribution,

$$Q_{j+1}(\theta|x) = p(\theta|x)\left[\frac{Q_j(\cdot|x)\,\bar{*}\,\kappa}{p(\cdot|x)\,\bar{*}\,\kappa} * \kappa\right](g^\theta). \tag{11}$$

The "$*$" symbol denotes group convolution and "$\bar{*}$" the combination of marginalization and group convolution (see App. A.4 for details). The true posterior $p(\theta|x)$ is clearly a fixed point of this sequence, with the number of iterations to convergence determined by $\kappa$ and the accuracy of the initialization network $q_{\text{init}}$.

---

[3]In practice, this results only in *approximate* samples due to the asymptotic behaviour of Gibbs sampling, and a potential mismatch between the trained $q$ and the targeted true posterior.

[4]This also explains why introducing the pose proxy is needed at all: GNPE would not work without it!

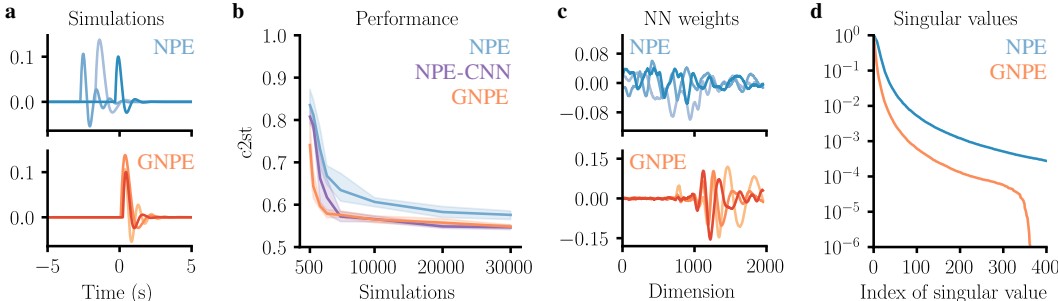

Figure 3: Comparison of standard NPE (blue) and GNPE (orange) for the damped harmonic oscillator. **a)** Three sample inputs to the neural density estimators showing GNPE data are pose-standardized. **b)** c2st score performance (best: 0.5, worst: 1.0): GNPE significantly outperforms equivariance-agnostic NPE, and is on par with NPE with a convolutional embedding network (purple). **c)** Example filters from the first linear layer of fully trained networks. GNPE filters more clearly capture oscillatory modes. **d)** Singular values of the training data. Inputs to GNPE $x'$ have smaller effective dimension than raw inputs $x$ to NPE.

## 4 TOY EXAMPLE: DAMPED HARMONIC OSCILLATOR

We now apply GNPE to invert a simple model of a damped harmonic oscillator. The forward model gives the time-dependent position $x$ of the oscillator, conditional on its real frequency $\omega_0$, damping ratio $\beta$, and time of excitation $\tau$. The time series $x$ is therefore a damped sinusoid starting at $\tau$ (and zero before). Noise is introduced via a normally-distributed perturbation of the parameters $\theta = (\omega_0, \beta, \tau)$, resulting in a Gaussian posterior $p(\theta|x)$ (further details in App. C.1). The model is constructed such that the posterior is equivariant under translations in $\tau$,

$$p(\omega_0, \beta, \tau + \Delta\tau | T_{\Delta\tau} x) = p(\omega_0, \beta, \tau | x), \tag{12}$$

so we take $\tau$ to be the pose. The equivariance of this model is exact, but it could easily be made approximate by, e.g., introducing $\tau$-dependent noise. The prior $p(\tau)$ extends from $-5$ s to $0$ s, so for NPE, the density estimator must learn to interpret data from oscillators excited throughout this range.

For GNPE, we shift the data to align the pose near $\tau = 0$ using a Gaussian kernel $\kappa = \mathcal{N}[0, (0.1 \text{ s})^2]$ (Fig. 3a). We then train a neural density estimator $q(\theta'|x')$ to approximate $p(\theta'|x')$, where $\theta' \equiv (\omega_0, \beta, -\epsilon)$ and $x' \equiv T_{-(\tau+\epsilon)} x$ are pose-standardized. We take $q(\theta'|x')$ to be diagonal Gaussian, matching the known form of the posterior. For each experiment, we train until the validation loss stops decreasing. We also train a neural density estimator $q_{\text{init}}(\tau|x)$ with standard NPE on the same dataset to generate initial GNPE seeds. To generate $N$ posterior samples we proceed as follows:

1. Sample $\tau^{(i)} \sim q_{\text{init}}(\tau|x)$, $i = 1, \ldots, N$;
2. Sample $\epsilon^{(i)} \sim \kappa(\epsilon)$, set $\hat{\tau}^{(i)} = \tau^{(i)} + \epsilon^{(i)}$, and time-translate the data, $x'^{(i)} = T_{-\hat{\tau}^{(i)}} x$;
3. Sample $\theta'^{(i)} \sim q(\theta'|x'^{(i)})$, and undo the time translation $\hat{\tau}^{(i)}$ to obtain $\theta^{(i)}$.

We repeat steps 2 and 3 until the distribution over $\tau$ converges. For this toy example and our choice of $\kappa$ only one iteration is required. For further details of the implementation see App. C.2.

We evaluate GNPE on five simulations by comparing inferred samples to ground-truth posteriors using the c2st score (Friedman, 2004; Lopez-Paz & Oquab, 2017). This corresponds to the test accuracy of a classifier trained to discriminate samples from the target and inferred distributions, and ranges from $0.5$ (best) to $1.0$ (worst). As baselines we evaluate standard NPE (i) with a network architecture identical to GNPE and (ii) with a convolutional embedding network (NPE-CNN; see App. C.2). Both approaches that leverage the equivariance, GNPE (by standardizing the pose) and NPE-CNN (by using a translation-equivariant embedding network), perform similarly well and far outperform standard NPE (Fig. 3b). This underscores the importance of equivariance awareness. The fact that the NPE network is trained to interpret signals from oscillators excited at arbitrary $\tau$, whereas GNPE focuses on signals starting around $\tau = 0$ (up to a small $\epsilon$ perturbation) is also reflected in in simplified filters in the first layer of the network (Fig. 3c) and a reduced effective dimension of the input data to GNPE (Fig. 3d).

## 5 GRAVITATIONAL-WAVE PARAMETER INFERENCE

Gravitational waves—propagating ripples of space and time—were first detected in 2015, from the inspiral, merger, and ringdown of a pair of black holes (Abbott et al., 2016). Since that time, the two LIGO detectors (Hanford and Livingston) (Aasi et al., 2015) as well as the Virgo detector (Acernese et al., 2015) have observed signals from over 50 coalescences of compact binaries involving either black holes or neutron stars (Abbott et al., 2019; 2021a;d). Key scientific results from these observations have included measurements of the properties of stellar-origin black holes that have provided new insights into their origin and evolution (Abbott et al., 2021b); an independent measurement of the local expansion rate of the Universe, the Hubble constant (Abbott et al., 2017); and new constraints on the properties of gravity and matter under extreme conditions (Abbott et al., 2018; 2021c).

Quasicircular binary black hole (BBH) systems are characterized by 15 parameters $\theta$, including the component masses and spins, as well as the space-time position and orientation of the system (Tab. D.1). Given these parameters, Einstein's theory of general relativity predicts the motion and emitted gravitational radiation of the binary. The GWs propagate across billions of light-years to Earth, where they produce a time-series signal $h_I(\theta)$ in each of the LIGO/Virgo interferometers $I = \mathrm{H, L, V}$. The signals on Earth are very weak and embedded in detector noise $n_I$. In part to have a tractable likelihood, the noise is approximated as additive and stationary Gaussian. The signal and noise models give rise to a likelihood $p(x|\theta)$ for observed data $x = \{h_I(\theta) + n_I\}_{I=\mathrm{H,L,V}}$.

Once the LIGO/Virgo detection pipelines are triggered, classical stochastic samplers are typically employed to determine the parameters of the progenitor system using Bayesian inference (Veitch et al., 2015; Ashton et al., 2019). However, these methods require millions of likelihood evaluations (and hence expensive waveform simulations) for each event analyzed. Even using fast waveform models, it can take $O(\mathrm{day})$ to analyze a single BBH. Faster inference methods are therefore highly desirable to cope with growing event rates, more realistic (and expensive) waveform models, and to make rapid localization predictions for possible multimessenger counterparts. Rapid amortized methods such as NPE have the potential to transform GW data analysis. However, due to the complexity and high dimensionality[5] of GW data, it has been a challenge (Gabbard et al., 2019; Chua & Vallisneri, 2020; Green & Gair, 2021; Delaunoy et al., 2020) to obtain results of comparable accuracy and completeness to classical samplers. We now show how GNPE can be used to exploit equivariances to greatly simplify the inference problem and achieve for the first time performance indistinguishable from "ground truth" stochastic samplers—at drastically reduced inference times.

### 5.1 EQUIVARIANCES OF SKY POSITION AND COALESCENCE TIME

We consider the analysis of BBH systems. Included among the parameters $\theta$ are the time of coalescence $t_c$ (as measured at geocenter) and the sky position (right ascension $\alpha$, declination $\delta$). Since GWs propagate at the speed of light, these are related to the times of arrival $t_I$ of the signal in each of the interferometers.[6] Our priors (based on the precision of detection pipelines) constrain $t_I$ to a range of $\approx 20$ ms, which is much wider than typical posteriors. Standard NPE inference networks must therefore be trained on simulations with substantial time shifts.

The detector coalescence times $t_I$—equivalently, $(t_c, \alpha, \delta)$—can alternatively be interpreted as the pose of the data, and standardized using GNPE. The group $G$ transforming the pose factorizes into a direct product of absolute and relative time shifts,

$$G = G_{\mathrm{abs}} \times G_{\mathrm{rel}}. \tag{13}$$

Group elements $g_{\mathrm{abs}} \in G_{\mathrm{abs}}$ act by uniform translation of all $t_I$, whereas $g_{\mathrm{rel}} \in G_{\mathrm{rel}}$ act by individual translation of $t_{\mathrm{L}}$ and $t_{\mathrm{V}}$. We work with data in frequency domain, where time translations act by multiplication, $T_g x_I = e^{-2\pi i f \Delta t_I} x_I$. Absolute time shifts correspond to a shift in $t_c$, and are an *exact* equivariance of the posterior, $p(g_{\mathrm{abs}}\theta|T_{g_{\mathrm{abs}}}x) = p(\theta|x)$. Relative time shifts correspond to a change in $(\alpha, \delta)$ (as well as $t_c$). This is only an *approximate* equivariance, since a change in sky position changes the projection of the incident signal onto the detector arms, leading to a subdominant change to the signal morphology in each detector.

---

[5]In our work, we analyze 8 s data segments between 20 Hz and 1024 Hz. Including also noise information, this results in 24,099 input dimensions per detector.

[6]We consider observations made in either $n_I = 2$ or 3 interferometers.

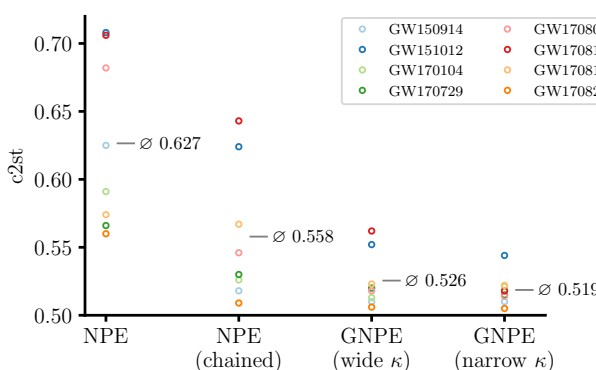

Figure 4: Comparison of estimated posteriors against LALINFERENCE MCMC for eight GW events, as quantified by c2st (best: 0.50, worst: 1.00). GNPE with a wide kernel outperforms both NPE baselines, while being only marginally slower (1 iteration $\sim 2$ s). With a narrow kernel and 30 iterations ($\sim 60$ s), we achieve c2st $< 0.55$ across all events. $\varnothing$ indicates the average across all eight events. For an alternative metric (MSE) see Fig. D.2.

## 5.2 APPLICATION OF GNPE

We use GNPE to standardize the pose within a band around $t_I = 0$. We consider two modes defined by different uniform blurring kernels. The "accurate" mode uses a narrow kernel $\kappa_{\text{narrow}} = U[-1 \text{ ms}, 1 \text{ ms}]^{n_I}$, whereas the "fast" mode uses a wide kernel $\kappa_{\text{wide}} = U[-3 \text{ ms}, 3 \text{ ms}]^{n_I}$. The latter is intended to converge in just one GNPE iteration, at the cost of having to interpret a wider range of data.

We define the blurred pose proxy $\hat{g}_I \equiv t_I + \epsilon_I$, where $\epsilon_I \sim \kappa(\epsilon_I)$. We then train a conditional density estimator $q(\theta'|x', \hat{g}_{\text{rel}})$, where $\theta' = \hat{g}_{\text{abs}}^{-1}\theta$ and $x' = T_{\hat{g}^{-1}}x$. That is, we condition $q$ on the relative time shift (since this is an approximate equivariance) and we translate parameters by the absolute time shift (since this is an exact equivariance). We always transform the data by the full time shift. We train a density estimator $q_{\text{init}}(\{t_I\}_{I=\text{H,L,V}}|x)$ using standard NPE to infer initial pose estimates.

The difficulty of the inference problem (high data dimensionality, significant noise levels, complex forward model) combined with high accuracy requirements to be scientifically useful requires careful design decisions. In particular, we initialize the first layer of the embedding network with principal components of clean waveforms to provide an inductive bias to extract useful information. We further use an expressive neural-spline normalizing flow (Durkan et al., 2019) to model the complicated GW posterior structure. See App. D.2 for details of network architecture and training.

## 5.3 RESULTS

We evaluate performance on all eight BBH events from the first Gravitational-Wave Transient Catalog (Abbott et al., 2019) consistent with our prior (component masses greater than 10 $M_\odot$). We generate reference posteriors with the LIGO/Virgo MCMC code LALINFERENCE (Veitch et al., 2015). We quantify the deviation between NPE samples and the reference samples using c2st.

We compare performance against two baselines, standard NPE and a modified approach that partially standardizes the pose ("chained NPE"). For the latter, we use the chain rule to decompose the posterior,

$$p(\theta|x) = p(\phi, \lambda|x) = p(\phi|x, \lambda) \cdot p(\lambda|x), \tag{14}$$

where $\lambda = (t_c, \alpha, \delta)$ are the pose parameters and $\phi \subset \theta$ collects the remaining 12 parameters. We use standard NPE to train a flow $q(\lambda|x)$ to estimate $p(\lambda|x)$, and a flow $q(\phi|x', \lambda)$ to estimate $p(\phi|x, \lambda)$. The latter flow is conditioned on $\lambda$, which we use to standardize the pose of $x$. In contrast to GNPE, this approach is sensitive to the initial pose estimate $q(\lambda|x)$, which limits accuracy (Figs. 4 and D.7). We note that all hyperparameters of the flow and training protocol (see App. D.2) were extensively optimized on NPE, and then transferred to GNPE without modification, resulting in conservative estimates of the performance advantage of GNPE. Fast-mode GNPE converges in one iteration, whereas accurate-mode requires 30 (convergence is assessed by the JS divergence between the inferred pose posteriors from two successive iterations).

Standard NPE performs well on some GW events but lacks the required accuracy for most of them, with c2st scores up to 0.71 (Fig. 4). Chained NPE performs better across the dataset, but performs poorly on events such as GW170814, for which the initial pose estimate is inaccurate. Indeed, we find

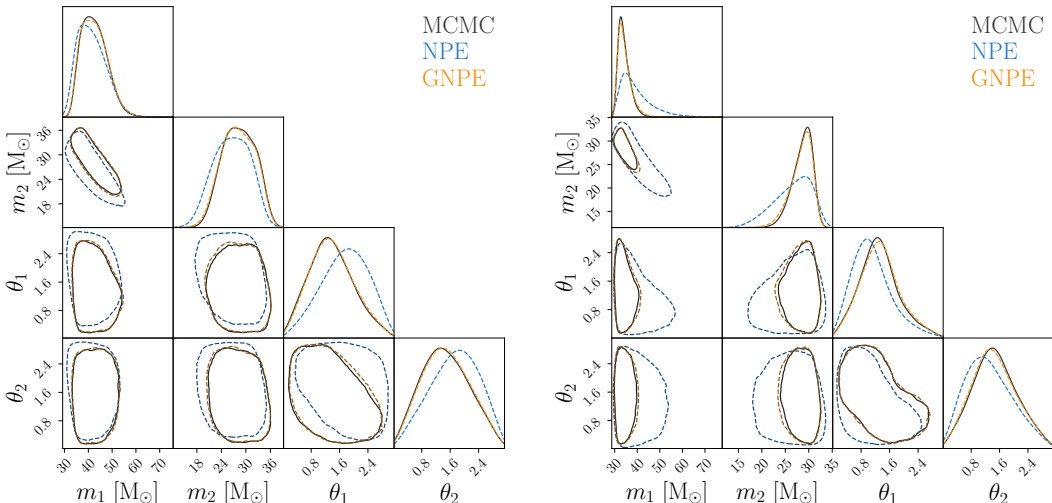

Figure 5: Corner plots for the GW events GW170809 (left) and GW170814 (right), plotting 1D marginals on the diagonal and 90% credible regions for the 2D correlations. We display the two black hole masses $m_1$ and $m_2$ and two spin parameters $\theta_1$ and $\theta_2$ (note that the full posterior is 15-dimensional). NPE does not accurately reproduce the MCMC posterior, while accurate-mode GNPE matches the MCMC results well. For a plot with all baselines see Fig. D.4.

that inaccuracies of that baseline can be almost entirely attributed to the initial pose estimate (Fig. D.6). Fast-mode GNPE with only one iteration is already more robust to this effect due to the blurring operation of the pose proxy (Fig. D.7). Both GNPE models significantly outperform the baselines, with accurate-mode obtaining c2st scores $< 0.55$ across all eight events. We emphasize that the c2st score is sensitive to any deviation between reference samples and samples from the inferred posterior. On a recent benchmark by Lueckmann et al. (2021) on examples with much lower parameter *and* data dimensions, even state-of-the-art SBI algorithms rarely reached c2st scores below $0.6$. The fact that GNPE achieves scores around $0.52$—i.e., posteriors which are nearly indistinguishable from the reference—on this challenging, high-dimensional, real-world example underscores the power of exploiting equivariances with GNPE.

Finally, we visualize posteriors for two events, GW170809 and GW170814, in Fig. 5. The quantitative agreement between GNPE and MCMC (Fig. 4) is visible from the overlapping marginals for all parameters displayed. NPE, by contrast, deviates significantly from MCMC in terms of shape and position. Note that we show a failure case of NPE here; for other events, such as GW170823, deviations of NPE from the reference posterior are less clearly visible.

## 6    CONCLUSIONS

We described GNPE, an approach to incorporate exact—and even *approximate*—equivariances under joint transformations of data and parameters into simulation-based inference. GNPE can be applied to black-box scientific forward models and any inference network architecture. It requires similar training times compared to NPE, while the added complexity at inference time depends on the number of GNPE iterations (adjustable, but typically $O(10)$). We show with two examples that exploiting equivariances with GNPE can yield large gains in simulation efficiency and accuracy.

For the motivating problem of GW parameter estimation, GNPE achieves for the first time rapid amortized inference with results virtually indistinguishable from MCMC (Dax et al., 2021). This is an extremely challenging "real-world" scientific problem, with high-dimensional input data, complex signals, and significant noise levels. It combines exact and approximate equivariances, and there is no clear path to success without their inclusion along with GW-specialized architectures and expressive density estimators.

## ETHICS STATEMENT

Our method is primarily targeted at scientific applications, and we do not foresee direct applications which are ethically problematic. In the context of GW analysis, we hope that GNPE contributes to reducing the required amount of compute, in particular when the rate of detections increases with more sensitive detectors in the future.

## REPRODUCIBILITY STATEMENT

The experimental setup for the toy model is described in App. C.2. We also provide the code at https://tinyurl.com/wmbjajv8. The setup for GW parameter inference is described in App. D.2. We will release a python package including scripts for the experiments carried out in section 5.

## ACKNOWLEDGMENTS

We thank A. Buonanno, T. Gebhard, J.M. Lückmann, S. Ossokine, M. Pürrer and C. Simpson for helpful discussions. We thank the anonymous reviewer for coming up with the illustration of GNPE in App. B. This research has made use of data, software and/or web tools obtained from the Gravitational Wave Open Science Center (https://www.gw-openscience.org/), a service of LIGO Laboratory, the LIGO Scientific Collaboration and the Virgo Collaboration. LIGO Laboratory and Advanced LIGO are funded by the United States National Science Foundation (NSF) as well as the Science and Technology Facilities Council (STFC) of the United Kingdom, the Max-Planck-Society (MPS), and the State of Niedersachsen/Germany for support of the construction of Advanced LIGO and construction and operation of the GEO600 detector. Additional support for Advanced LIGO was provided by the Australian Research Council. Virgo is funded, through the European Gravitational Observatory (EGO), by the French Centre National de Recherche Scientifique (CNRS), the Italian Istituto Nazionale di Fisica Nucleare (INFN) and the Dutch Nikhef, with contributions by institutions from Belgium, Germany, Greece, Hungary, Ireland, Japan, Monaco, Poland, Portugal, Spain. This material is based upon work supported by NSF's LIGO Laboratory which is a major facility fully funded by the National Science Foundation. M. Dax thanks the Hector Fellow Academy for support. M. Deistler thanks the International Max Planck Research School for Intelligent Systems (IMPRS-IS) for support. B.S. and J.H.M. are members of the MLCoE, EXC number 2064/1 – Project number 390727645. This work was supported by the German Federal Ministry of Education and Research (BMBF): Tübingen AI Center, FKZ: 01IS18039A. We use `PyTorch` (Paszke et al., 2019), `nflows` (Durkan et al., 2020) and `sbi` (Tejero-Cantero et al., 2020) for the implementation of our neural networks. The plots are generated with `matplotlib` (Hunter, 2007) and `ChainConsumer` (Hinton, 2016).

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

# A   DERIVATIONS

## A.1   EQUIVARIANCE RELATIONS

Consider a system with an exact equivariance under a joint transformation of parameters $\theta$ and observations $x$,

$$\theta \to \theta' = g\theta, \tag{15}$$

$$x \to x' = T_g x. \tag{16}$$

An invariant prior fulfills the relation

$$p(\theta) = p(\theta') \left| \det J_g \right|, \tag{17}$$

where the Jacobian $J_g$ arises from the change of variables rule for probability distributions. A similar relation holds for an equivariant likelihood,

$$p(x|\theta) = p(x'|\theta') \left| \det J_T \right|. \tag{18}$$

An invariant prior and an equivariant likelihood further imply for the evidence $p(x)$

$$p(x) = \int p(x|\theta)p(\theta)d\theta = \int p(x'|\theta') \left| \det J_T \right| p(\theta') \left| \det J_g \right| \, d\theta = p(x') \left| \det J_T \right|. \tag{19}$$

Combining an invariant prior with an equivariant likelihood thus leads to the equivariance relation

$$\begin{aligned} p(\theta|x) = \frac{p(x|\theta)p(\theta)}{p(x)} &= \frac{p(x'|\theta') \left| \det J_T \right| p(\theta') \left| \det J_g \right|}{p(x') \left| \det J_T \right|} \\ &= p(\theta'|x') \left| \det J_g \right| \end{aligned} \tag{20}$$

for the posterior, where we used Bayes' theorem and equations (17), (18), and (19).

## A.2   EQUIVARIANCE OF $p(\theta|x, \hat{g})$

Here we derive that an equivariant posterior $p(\theta|x)$ remains equivariant if the distribution is also conditioned on the proxy $\hat{g}$, as used in equation (7). With the definition $p(\hat{g}|\theta) = \kappa \left( (g^\theta)^{-1} \hat{g} \right)$ from section 3.3, $p(\hat{g}|\theta)$ is equivariant under joint application of $h \in G$ to $\hat{g}$ and $\theta$,

$$p(\hat{g}|\theta) = \kappa \left( (g^\theta)^{-1} \hat{g} \right) = \kappa \left( (g^\theta)^{-1} h^{-1} h \hat{g} \right) = \kappa \left( (hg^\theta)^{-1} h\hat{g} \right) = p(h\hat{g}|h\theta). \tag{21}$$

where for the last equality we used $g^{h\theta} = hg^\theta$. This implies, that $p(\hat{g}|x)$ is equivariant under joint application of $h$ and $T_h$,

$$\begin{aligned} p(\hat{g}|x) = \int p(\hat{g}|\theta, x)p(\theta|x) \, d\theta &\stackrel{(21),(4)}{=} \int p(h\hat{g}|h\theta)p(h\theta|T_h x) \left| \det J_h \right| \, d\theta \\ &= p(h\hat{g}|T_h x). \end{aligned} \tag{22}$$

in the second step we used $p(\hat{g}|\theta, x) = p(\hat{g}|\theta)$. From these relations, the equivariance relation used in equation (7) follows,

$$\begin{aligned} p(\theta|x, \hat{g}) = \frac{p(\theta, \hat{g}|x)}{p(\hat{g}|x)} = \frac{p(\hat{g}|\theta, x)p(\theta|x)}{p(\hat{g}|x)} &\stackrel{(21),(4),(22)}{=} \frac{p(h\hat{g}|h\theta)p(h\theta|T_h x)}{p(h\hat{g}|T_h x)} \left| \det J_h \right| \\ &= p(h\theta|T_h x, h\hat{g}) \left| \det J_h \right|. \end{aligned} \tag{23}$$

## A.3   EXACT EQUIVARIANCE OF INFERRED POSTERIOR

Consider a posterior that is exactly equivariant under $G$,

$$p(\theta|x) = p(g\theta|T_g x) |\det J_g|, \qquad \forall g \in G. \tag{24}$$

We here show that the posterior estimated using GNPE is equivariant under $G$ by construction. This holds regardless of whether $q(\theta'|x')$ has fully converged to $p(\theta'|x')$.

With GNPE, the equivariant posterior $p(\theta|x_o)$ for an observation $x_o$ is inferred by alternately sampling

$$\theta^{(i)} \sim p(\theta|x_o, \hat{g}^{(i-1)}) \qquad \Longleftrightarrow \qquad \theta^{(i)} = \hat{g}^{(i-1)}\theta'^{(i)}, \quad \theta'^{(i)} \sim q(\theta'|T_{(\hat{g}^{(i-1)})^{-1}}x_o), \qquad (25)$$

$$\hat{g}^{(i)} \sim p(\hat{g}|x_o, \theta^{(i)}) \qquad \Longleftrightarrow \qquad \hat{g}^{(i)} = g^{\theta^{(i)}}\epsilon, \quad \epsilon \sim \kappa(\epsilon), \qquad (26)$$

see also equation (9). Now consider a different observation $\tilde{x}_o = T_h x_o$ that is obtained by altering the pose of $x_o$ with $T_h$, where $h$ is an arbitrary element of the equivariance group $G$. Applying the joint transformation

$$\hat{g} \to h\hat{g}, \qquad (27)$$
$$x_o \to T_h x_o, \qquad (28)$$

in (25) leaves $\theta'$ invariant,

$$q(\theta'|T_{(h\hat{g})^{-1}}T_h x_o) = q(\theta'|T_{\hat{g}^{-1}}(T_h)^{-1}T_h x_o) = q(\theta'|T_{\hat{g}^{-1}}x_o). \qquad (29)$$

We thus find that $\theta$ in (25) transforms equivariantly under joint application of (27) and (28),

$$\theta = \hat{g}\theta' \to (h\hat{g})\theta' = h(\hat{g}\theta') = h\theta. \qquad (30)$$

Conversely, applying

$$\theta \to h\theta \qquad (31)$$

in (26) transforms $\hat{g}$ by

$$\hat{g} = g^\theta \epsilon \to g^{(h\theta)}\epsilon = hg^\theta \epsilon = h\hat{g}. \qquad (32)$$

The $\theta$ samples (obtained by marginalizing over $\hat{g}$) thus transform $\theta \to h\theta$ under $x_o \to T_h x_0$, which is consistent with the desired equivariance (24).

Another intuitive way to see this is to consider running an implementation of the Gibbs sampling steps (25) and (26) with fixed random seed for two observations $x_o$ (initialized with $\hat{g}^{(0)} = \hat{g}_{x_o}$) and $T_h x_o$ (initialized with $\hat{g}^{(0)} = h\hat{g}_{x_o}$). The Gibbs sampler will yield parameters samples $(\theta_i)_{i=1}^N$ for $x_o$, and the *exact same* samples $(h\theta_i)_{i=1}^N$ for $T_h x_o$, up to the global transformation by $h$. The reason is that the density estimator $q(\theta'|x')$ is queried with the same $x'$ for both observations $x_o$ and $T_h x_o$ in each iteration $i$. Since the truncated, thinned samples are asymptotically independent of the initialization, this shows that (24) is fulfilled by construction.

## A.4 ITERATIVE INFERENCE AND CONVERGENCE

GNPE leverages a neural density estimator of the form $q(\theta|x', \hat{g})$ to obtain samples from the joint distribution $p(\theta, \hat{g}|x)$. This is done by iterative sampling as described in section 3. Here we derive equation (11), which states how a distribution $Q_j(\theta|x)$ is updated by a single GNPE iteration.

Given a distribution $Q_j^\theta(\theta|x)$ of $\theta$ samples in iteration $j$, we infer samples for the pose proxy $\hat{g}$ for the next iteration by (i) extracting the pose $g^\theta$ from $\theta$ (this essentially involves marginalizing over all non pose related parameters) and (ii) blurring the pose $g^\theta$ with the kernel $\kappa$, corresponding to a group convolution.[A.1] We denote this combination of marginalization and group convolution with the "$\bar{\ast}$" symbol,

$$Q_{j+1}^{\hat{g}}(\hat{g}|x) = \int d\theta\, Q_j^\theta(\theta|x)\kappa((g^\theta)^{-1}\hat{g}) = \left(Q_j^\theta(\cdot|x) \bar{\ast} \kappa\right)(\hat{g}). \qquad (33)$$

For a given proxy sample $\hat{g}$, a (perfectly trained) neural density estimator infers $\theta$ with

$$p(\theta|x, \hat{g}) = \frac{p(\theta, \hat{g}|x)}{p(\hat{g}|x)} = \frac{p(\hat{g}|x, \theta)p(\theta|x)}{p(\hat{g}|x)} = p(\theta|x)\frac{\kappa((g^\theta)^{-1}\hat{g})}{(p^\theta(\cdot|x) \bar{\ast} \kappa)(\hat{g})}, \qquad (34)$$

---

[A.1]We define a group convolution as $(A \ast B)(\hat{g}) = \int dg\, A(g)B(g^{-1}\hat{g})$, which is the natural extension of a standard convolution.

where we used $p(\hat{g}|\theta) = \kappa((g^\theta)^{-1}\hat{g})$. Combining (33) and (34), the updated distribution over $\theta$ samples reads

$$
\begin{aligned}
Q^\theta_{j+1}(\theta|x) &= \int d\hat{g}\, p(\theta|x,\hat{g})\, Q^{\hat{g}}_{j+1}(\hat{g}|x) \\
&= \int d\hat{g}\, \left(Q^\theta_j(\cdot|x)\,\bar{\divideontimes}\,\kappa\right)(\hat{g})\, p(\theta|x)\frac{\kappa((g^\theta)^{-1}\hat{g})}{((p^\theta(\cdot|x)\,\bar{\divideontimes}\,\kappa))(\hat{g})} \\
&= p(\theta|x) \int d\hat{g}\, \frac{\left(Q^\theta_j(\cdot|x)\,\bar{\divideontimes}\,\kappa\right)(\hat{g})}{((p^\theta(\cdot|x)\,\bar{\divideontimes}\,\kappa))(\hat{g})}\,\kappa((g^\theta)^{-1}\hat{g}) \\
&= p(\theta|x) \left(\frac{Q^\theta_j(\cdot|x)\,\bar{\divideontimes}\,\kappa}{p^\theta(\cdot|x)\,\bar{\divideontimes}\,\kappa}\ *\ \kappa^{(-)}\right)(\hat{g}).
\end{aligned}
\tag{35}
$$

Here, $\kappa^{(-)}$ denotes the reflected kernel, $\kappa^{(-)}(g) = \kappa(g^{-1})\,\forall g$. Since we choose a symmetric kernel in practice, we use $\kappa = \kappa^{(-)}$ in (11).

In this notation, the initialization of the pose $g^\theta$ in iteration 0 with $q_{\text{init}}$ simply means setting $Q_0(\cdot|x)\,\bar{\divideontimes}\,\kappa = q_{\text{init}}(\cdot|x)*\kappa$.

## B    GNPE FOR SIMPLE GAUSSIAN LIKELIHOOD AND PRIOR

Consider a simple forward model $\tau \to x$ with a given prior
$$
p(\tau) = \mathcal{N}(-5,1)[\tau]
\tag{36}
$$
and a likelihood
$$
p(x|\tau) = \mathcal{N}(\tau,1)[x],
\tag{37}
$$
where the normal distribution is defined by
$$
\mathcal{N}(\mu,\sigma^2)[x] = \frac{\exp\left(\frac{-(x-\mu)^2}{2\sigma^2}\right)}{\sqrt{2\pi}\sigma}.
\tag{38}
$$
The evidence can be computed from the prior (36) and likelihood (37), and reads
$$
p(x) = \int d\tau p(\tau)p(x|\tau) = \int d\tau \mathcal{N}(-5,1)[\tau]\mathcal{N}(\tau,1)[x] = \mathcal{N}(-5,2)[x].
\tag{39}
$$
The posterior is then given via Bayes' theorem and reads
$$
p(\tau|x) = \frac{p(x|\tau)p(\tau)}{p(x)} = \frac{\mathcal{N}(\tau,1)[x]\mathcal{N}(-5,1)[\tau]}{\mathcal{N}(-5,\sqrt{2})[x]} = \mathcal{N}\left(\frac{x-5}{2},1/2\right)[\tau].
\tag{40}
$$

### B.1    EQUIVARIANCES

The likelihood (37) is equivariant under $G$, i.e., the joint transformation
$$
\begin{aligned}
\tau &\to g\tau = \tau + \Delta\tau, \\
x &\to T^l_g x = x + \Delta\tau.
\end{aligned}
\tag{41}
$$
This follows directly from (37) and (38). If the prior was invariant, then this equivariance would be inherited by the posterior, see App. A.1. However, the prior is not invariant. It turns out that the posterior is still equivariant, but $x$ transforms under a different representation than it does for the equivariance of the likelihood. Specifically, the posterior (40) is equivariant under joint transformation
$$
\begin{aligned}
\tau &\to g\tau = \tau + \Delta\tau, \\
x &\to T^p_g x = x + 2\cdot\Delta\tau,
\end{aligned}
\tag{42}
$$
which again directly follows from (40) and (38). Importantly, $T^p_g \neq T^l_g$, i.e., the representation under which $x$ transforms is different for the equivariance of the likelihood and the posterior. For GNPE, the relevant equivariance is that of the posterior, i.e. the set of transformations (42), see also equation (4). The equivariance relation of the posterior thus reads
$$
p(\tau|x) = p(g\tau|T^p_g x)|\det J_g|, \qquad \forall g\in G.
\tag{43}
$$

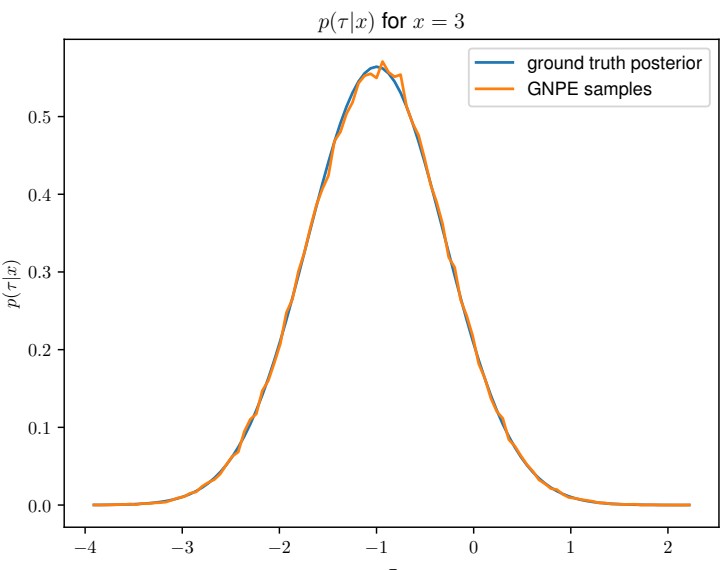

Figure B.1: Posterior $p(\tau|x = 3)$ (blue) and the corresponding inferred GNPE samples (orange).

## B.2 GNPE

We choose $\tau$ as the pose, which we aim to standardize with GNPE. We define the corresponding proxy as

$$\hat{\tau} = \tau + \epsilon, \quad \epsilon \sim \kappa(\epsilon) = \mathcal{N}(0,1)[\epsilon]. \tag{44}$$

We can use GNPE to incorporate the exact equivariance of the posterior by construction. To that end we define

$$\begin{aligned} \tau' &= g^{(-\hat{\tau})}\tau = \tau - \hat{\tau}, \\ x' &= T^p_{g^{(-\hat{\tau})}}x = x - 2 \cdot \hat{\tau}. \end{aligned} \tag{45}$$

We then train a neural density estimator to estimate $p(\tau'|x')$. This distribution is of the same form as $p(\tau|x)$ and simply given by

$$p(\tau'|x') \stackrel{(43),(40)}{=} \mathcal{N}\left(\frac{x'-5}{2}, 1/2\right)[\tau'] \tag{46}$$

due to the equivariance (43). We here assume a neural density estimator that estimates (46) perfectly. For GNPE, we

1. Initialize $\tau^{(1)} = 0$;
2. Sample $\hat{\tau}^{(1)}$ by $\hat{\tau}^{(1)} = \tau^{(1)} + \epsilon$, $\epsilon \sim \mathcal{N}(0,1)[\epsilon]$, and compute $\tau'$ and $x'$ via (45);
3. Sample $\tau^{(2)}$ by $\tau^{(2)} = \tau'^{(2)} + \hat{\tau}^{(1)}$, with $\tau'^{(2)} \sim p(\tau'|x') = \mathcal{N}\left(\frac{x'-5}{2}, 1/2\right)[\tau']$;

and repeat (2) and (3) multiple times. This constructs a Markov chain. To obtain (approximately independent) posterior samples $\tau \sim p(\tau|x)$, we truncate to account for burn-in, thin the chain and marginalize over $\hat{\tau}$. We find that the chain indeed converges to the correct posterior (40), see Fig. B.1.

## C TOY EXAMPLE

### C.1 FORWARD MODEL

The toy model in section 4 describes the motion of a damped harmonic oscillator that is initially at rest and excited at time $\tau$ with an infinitely short pulse. The time evolution of that system is governed

by the differential equation

$$\frac{d^2}{dt^2}x(t) + 2\beta\omega_0 \frac{d}{dt}x(t) + \omega_0^2 x(t) = \delta(t - \tau), \tag{47}$$

where $\omega_0$ denotes the undamped angular frequency and $\beta$ the damping ratio. The solution for the time series $x(t)$ is given by the Green's function for the corresponding differential operator and reads

$$x(t) = \begin{cases} 0, & t \leq \tau \\ e^{-\beta\omega_0(t-\tau)} \cdot \frac{\sin\left(\sqrt{1-\beta^2}\omega_0(t-\tau)\right)}{\sqrt{1-\beta^2}\omega_0}, & t > \tau. \end{cases} \tag{48}$$

This equation describes a deterministic, injective mapping between parameters $\theta = (\omega_0, \beta, \tau)$ and a time series observation $x$,

$$x = f(\theta). \tag{49}$$

This implies a likelihood $p(x|\theta) = \delta(x - f(\theta))$, and thus a point-like posterior. To showcase (G)NPE on this toy problem we introduce stochasticity by setting

$$x = f(\theta + \delta\theta) \tag{50}$$

instead. We sample $\delta\theta$ from an uncorrelated Gaussian distribution

$$\delta\theta \sim \mathcal{N}(0, \Sigma), \qquad \Sigma = \begin{pmatrix} \sigma_{\omega_0}^2 & 0 & 0 \\ 0 & \sigma_\beta^2 & 0 \\ 0 & 0 & \sigma_\tau^2 \end{pmatrix}, \tag{51}$$

with $\sigma_{\omega_0} = 0.3$ Hz, $\sigma_\beta = 0.03$ and $\sigma_\tau = 0.3$ s. Due to the injectivity of $f$, the posterior $p(\theta|x)$ reduces to the probability $p(\delta\theta = f^{-1}(x) - \theta)$. With a uniform prior, and neglecting boundary effects, this implies an uncorrelated Gaussian posterior $p(\theta|x)$ centered around $f^{-1}(x)$ with standard deviations as specified above. We choose this approach over, e.g., adding noise straight to observations to keep the problem as simple as possible, such that the focus remains on the comparison of GNPE and NPE. In particular, knowing that the ground truth posteriors are Gaussian, we can use a simple Gaussian density estimator.

We choose uniform priors

$$p(\omega_0) = \mathcal{U}[3, 10]\ \text{Hz}, \qquad p(\beta) = \mathcal{U}[0.2, 0.5], \qquad p(\tau) = \mathcal{U}[-5, 0]\ \text{s}. \tag{52}$$

The observational data $x$ is the discretized time series in the interval $[-5, +5]$ s with 2000 evenly sampled bins. When applying time shifts with GNPE, we impose cyclic boundary conditions.

## C.2 IMPLEMENTATION

We use a Gaussian density estimator for all methods (since we know that the true posterior is Gaussian). For NPE, we use a feedforward neural network with [128, 32, 16] hidden units and with ReLU activation functions as an embedding network. For NPE-CNN, we use a three-layer convolutional embedding network with kernel sizes [5,5,5], stride 1, [6,12,12] channels, average pooling with kernel size 7 and stride 7, and ReLU activation functions. For GNPE, we use the same architecture as for NPE for both, $q(\theta'|x')$ and $q_{\text{init}}(\tau|x)$. For further hyperparameters, we use the defaults of the sbi package (Tejero-Cantero et al., 2020).

## C.3 RESULTS

For all methods, we compute the average classifier two-sample test score (c2st) based on 10,000 samples from the estimated and the ground truth posterior for five different simulations. We then average the accuracy across 10 different seeds.

## D GRAVITATIONAL WAVE PARAMETER INFERENCE

### D.1 FORWARD MODEL AND AMORTIZATION

The forward model mapping binary black hole parameters $\theta$ (Tab. D.1) to simulated measurements $x$ in the detectors consists of two stages. Firstly, the waveform polarizations $h(\theta)$ for given parameters

Table D.1: Priors for the astrophysical binary black hole parameters used to train the inference network. Priors are uniform over the specified range unless indicated otherwise. We train networks with different distance ranges for the two observing runs O1 and O2 due to the different detector sensitivities. At inference time, a cosmological distance prior is imposed by reweighting samples according to their distance.

| Description | Parameter | Prior |
|---|---|---|
| component masses | $m_1, m_2$ | $[10, 80]$ $M_\odot$, $m_1 \geq m_2$ |
| spin magnitudes | $a_1, a_2$ | $[0, 0.88]$ |
| spin angles | $\theta_1, \theta_2, \phi_{12}, \phi_{JL}$ | standard as in Farr et al. (2014) |
| time of coalescence | $t_c$ | $[-0.1, 0.1]$ s |
| luminosity distance | $d_L$ | O1, 2 detectors: $[100, 2000]$ Mpc |
| | | O2, 2 detectors: $[100, 2000]$ Mpc and $[100, 6000]$ Mpc |
| | | O2, 3 detectors: $[100, 1000]$ Mpc |
| reference phase | $\phi_c$ | $[0, 2\pi]$ |
| inclination | $\theta_{JN}$ | $[0, \pi]$ uniform in sine |
| polarization | $\psi$ | $[0, \pi]$ |
| sky position | $\alpha, \beta$ | uniform over sky |

$\theta$ are computed with the waveform model IMRPhenomPv2 (Hannam et al., 2014; Khan et al., 2016; Bohé et al., 2016). Secondly, the signals are projected onto the detectors, and noise is added to obtain a realistic signal $x$. To a good approximation, we assume the noise to be Gaussian and stationary over the duration of a single GW signal. However, the noise spectrum, determined by the power spectral density (PSD) $S_n$, drifts over the duration of an observing run. To fully amortize the computational cost, we use a variety of different PSDs $S_n$ in training, and additionally condition the inference network on $S_n$. At inference time, this enables instant tuning of the inference network to the PSD estimated at the time of the event, see Dax et al. (2021) for details. Since this conditioning on $S_n$ has no effect on the GNPE algorithm outlined in this work, we keep it implicit in all equations.

## D.2 NETWORK ARCHITECTURE AND TRAINING

The inference network consists of an embedding network, that reduces the high dimensional input data to a 128 dimensional feature vector, and the normalizing flow, that takes this feature vector as input. For each detector, the input to the embedding network consists of the complex-valued frequency domain strain in the range $[20 \text{ Hz}, 1024 \text{ Hz}]$ with a resolution of $0.125$ Hz, and PSD information $(10^{46} \cdot S_n)^{-1/2}$ with the same binning. This results to a total of $(3 \cdot 8{,}033) = 24{,}099$ real input bins per detector. The first module of the embedding network consists of a linear layer per detector, that maps this $24{,}096$ dimensional input to $400$ components. We initialize this compression layer with PCA components of raw waveforms. This provides a strong inductive bias to the network to filter out GW signals from extremely noisy data. Note that this important step is only possible since GNPE is architecture independent—it is for instance not compatible with a convolutional neural network. Following this compression layer, we use a series of 24 fully-connected residual blocks with two layers each to compress the output to the desired 128 dimensional feature vector. We use batch normalization and ELU activation functions. Importantly, the conditioning of the flow on the proxy $\hat{g}_{\text{rel.}}$ is done *after* the embedding network, by concatenating $\hat{g}_{\text{rel.}}$ to the embedded feature vector.

Following this, we use a neural spline flow (Durkan et al., 2019) with rational-quadratic spline coupling transforms as density estimator. We use 30 such transforms, each of which is associated with 5 two-layer residual blocks with hidden dimension 512. In total, the inference network has 348 hidden layers and $1.31 \cdot 10^8$ (for two detectors) or $1.42 \cdot 10^8$ (for three detectors) learnable parameters.

We train the inference network with a data set of $5 \cdot 10^6$ waveforms with parameters $\theta$ sampled from the priors specified in table D.1, and reserve 2% of the data for validation. We pretrain the network with learning rate of $3 \cdot 10^{-4}$ for 300 epochs with fixed PSD, and finetune for another 150 epochs with learning rate of $3 \cdot 10^{-5}$ with varying PSDs. With batch size 4,096, training takes 16-18 days on a NVIDIA Tesla V100 GPU.

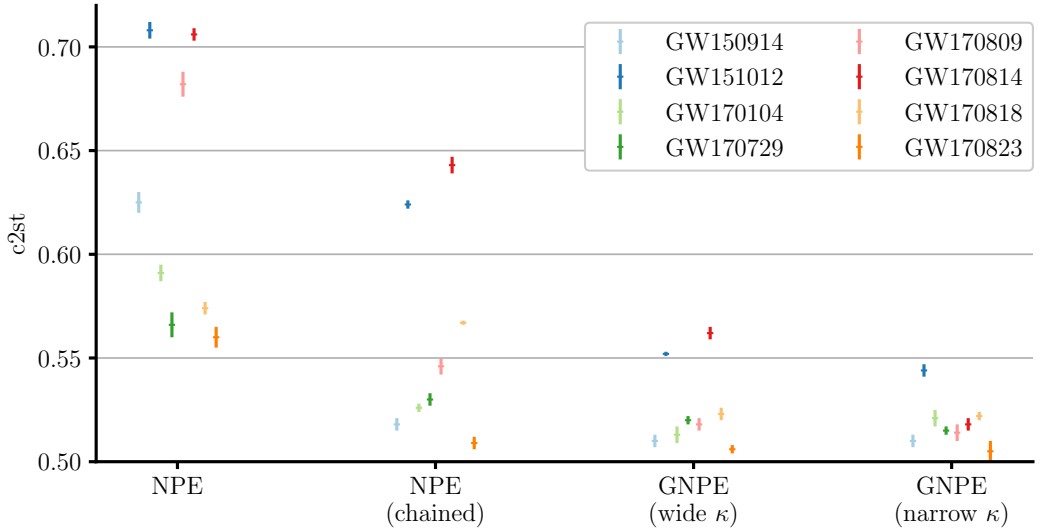

Figure D.1: c2st scores quantifying the deviation between the inferred posteriors and the MCMC reference. This is an extended version of Fig. 4.

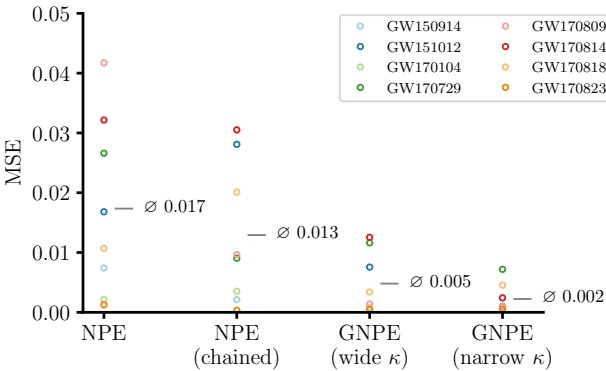

Figure D.2: Comparison of estimated posteriors against LALINFERENCE MCMC for eight GW events, as quantified by the mean squared error (MSE) of the sample means. Before computing the means, we normalize each dimension such that the prior has a standard deviation of 1. ∅ indicates the average across all eight events. GNPE with a narrow kernel consistently outperforms the baselines, which is in accordance with Fig. 4.

## D.3 RESULTS

The c2st scores between inferred posterior and the MCMC reference shown in Fig. 4 are computed using the code and default hyperparameters of Lueckmann et al. (2021). For each event, we compute the c2st score of 10,000 samples for inferred and target posterior. Fig. 4 displays the mean of the score across 5 different sample realizations, Fig. D.1 additionally shows the corresponding standard deviation. For technical reasons we use only 12 of the 15 inferred parameters; specifically we omit the geocentric time of coalescence $t_c$ (since the reference posteriors generated with LALInference do not contain that variable) and the sky position parameters $\alpha$ and $\delta$ (since the NPE baseline with chain rule decomposition infers these in another basis).

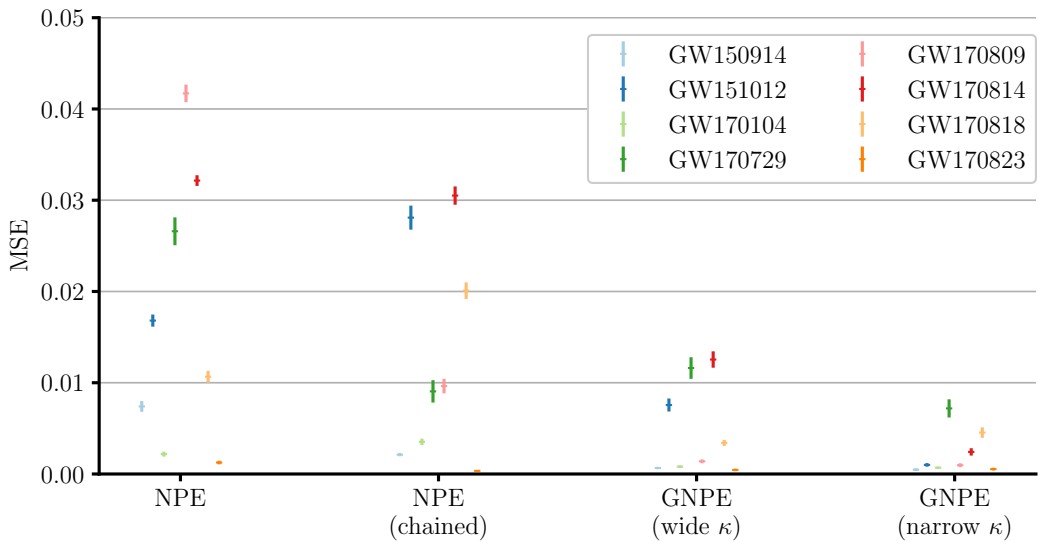

Figure D.3: MSE between inferred posteriors and MCMC reference. Extended version of Fig. D.2.

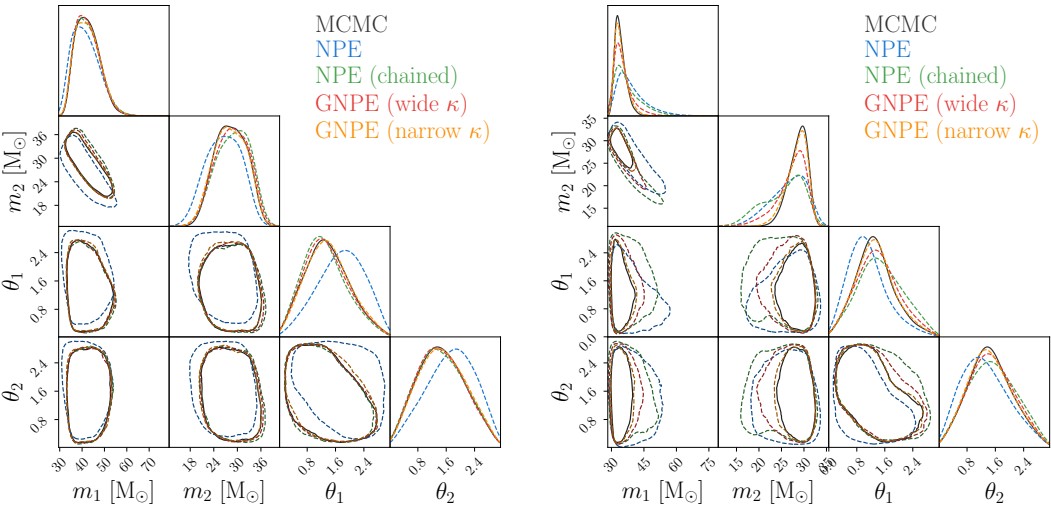

Figure D.4: Corner plots for the GW events GW170809 (left) and GW170814 (right), plotting 1D marginals on the diagonal and 90% credible regions for the 2D correlations. We display the two black hole masses $m_1$ and $m_2$ and two spin parameters $\theta_1$ and $\theta_2$ (note that the full posterior is 15-dimensional). This extends Fig. 5 by also displaying the results from chained NPE and GNPE with wide $\kappa$.

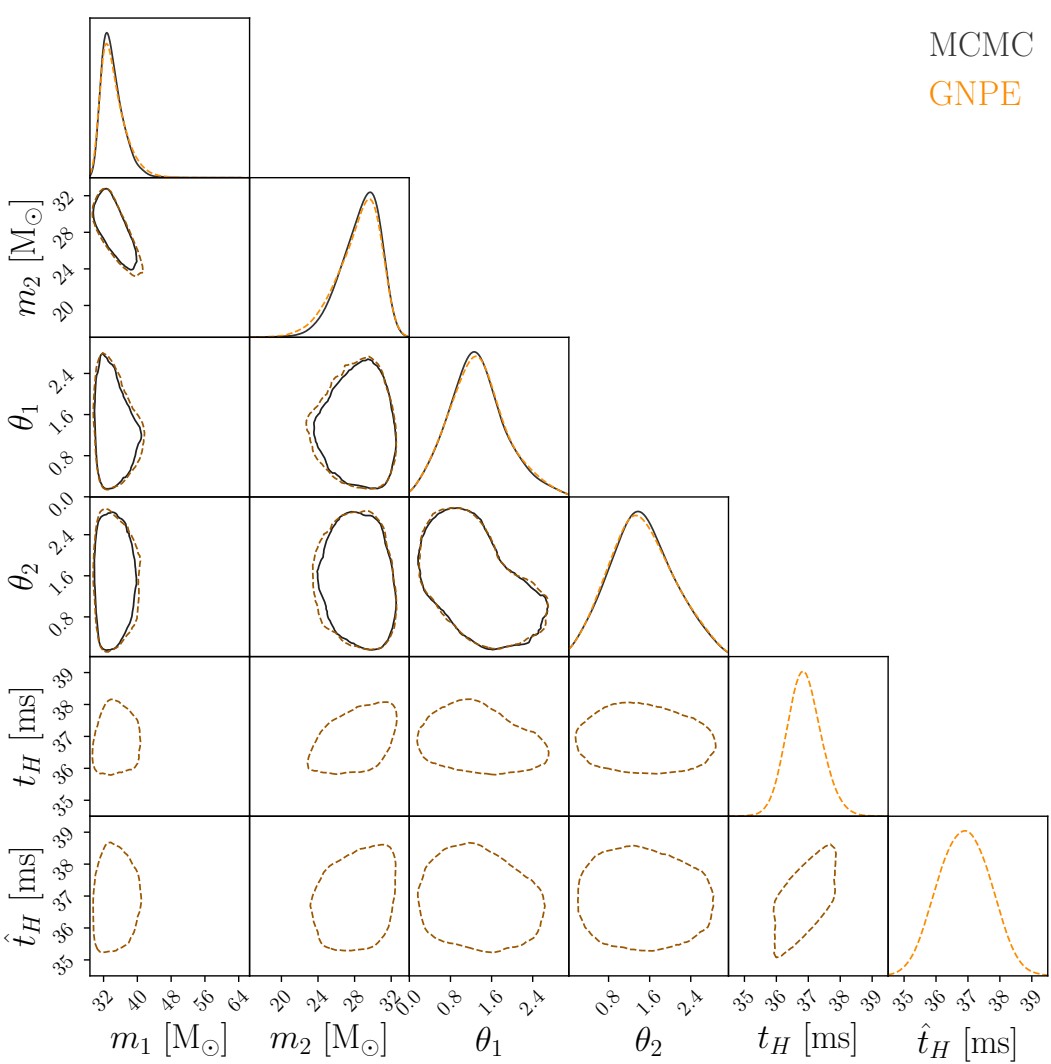

Figure D.5: Corner plot for the GW event GW170814 plotting 1D marginals on the diagonal and 90% credible regions for the 2D correlations. We display the two black hole masses $m_1$ and $m_2$ and two spin parameters $\theta_1$ and $\theta_2$ that are also shown in Fig. 5. We additionally display one of the pose parameters $t_H$ and the corresponding proxy $\hat{t}_H$ from the last GNPE iteration. In training, the neural density estimator learned that the true pose $t_H$ differs by at most 1 ms from the proxy $\hat{t}_H$ that it is conditioned on (since we chose a kernel $\kappa_{\text{narrow}} = U[-1 \text{ ms}, 1 \text{ ms}]^{n_I}$, see section 5.2). This explains the strong correlation between $t_H$ and $\hat{t}_H$ we observe. For the same reason, the observed correlations between the $\hat{t}_H$ and the non-pose parameters $(m_1, m_2, \theta_1, \theta_2)$ are similar to those between the true pose $t_H$ and $(m_1, m_2, \theta_1, \theta_2)$.

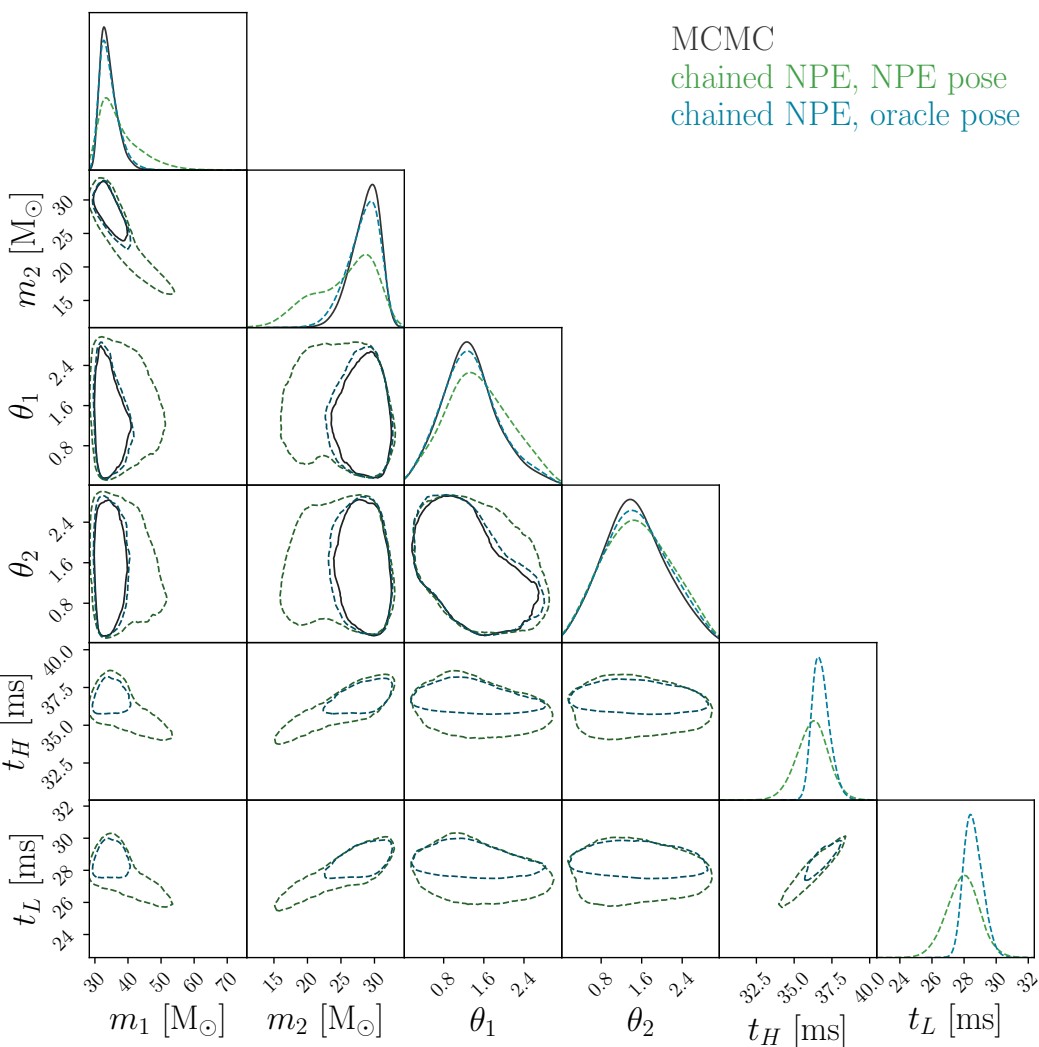

Figure D.6: Corner plot for GW170814 with the four parameters $(m_1, m_2, \theta_1, \theta_2)$ that are also displayed in Fig. 5, as well as the pose $(t_H, t_L)$. We compare chained NPE as described in section 5.3 to an oracle version: for the earlier the pose is inferred using standard NPE (green) while for the latter we take an oracle pose provided by a (slow) nested sampling algorithm (teal). We observe, that the result using the oracle pose matches the MCMC reference posterior well, while the other one shows clear deviations. Both versions use the same density estimator for the non-pose parameters $\phi \subset \theta$. This demonstrates that inaccuracies of the chained NPE baselines can be almost entirely attributed to inaccurate inital estimates of the pose. Poor pose estimates can occur since the density estimator trained to extract the pose operates on non pose-standardized data.

Note: The MCMC reference algorithm LALInference does not provide full pose information since it automatically marginalizes over $t_c$. For the oracle pose we thus employ the nested sampling algorithm bilby (Ashton et al., 2019).

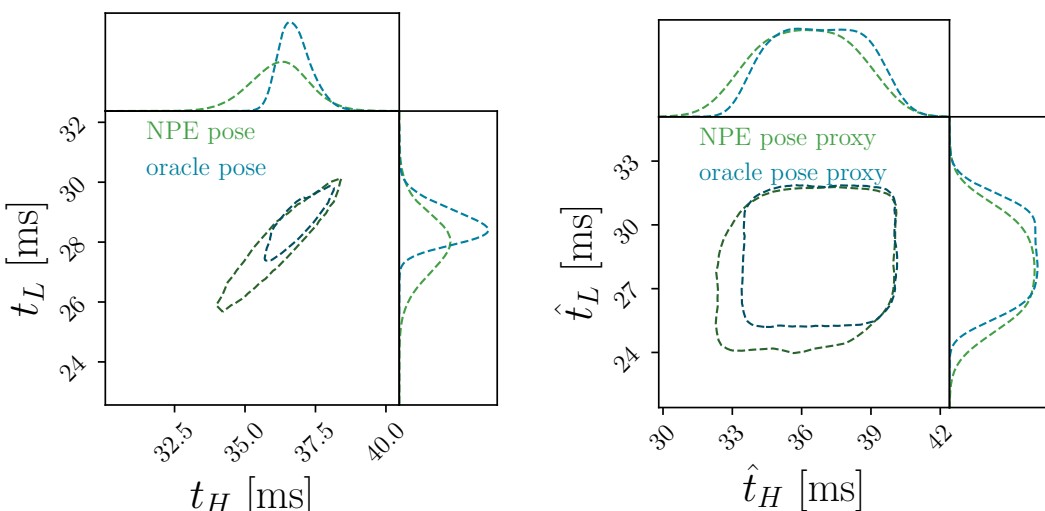

Figure D.7: Left: Pose parameters $t_H$ and $t_L$ for the GW event GW170814, estimated with the neural density estimator $q_{\text{init}}$ with standard NPE (green), as well as the "true" pose inferred with bilby (teal). Right: Pose proxies $\hat{t}_H$ and $\hat{t}_L$ for the wide kernel $\kappa_{\text{wide}} = U[-3 \text{ ms}, 3 \text{ ms}]^{n_I}$. These are obtained from the pose estimates in the left panel via a convolution with $\kappa_{\text{wide}}$. We observe that the deviation between the oracle and the NPE estimate is substantially smaller for the pose proxy than for the pose itself due to the blurring operation. This leads to a better performance of fast-mode GNPE (with $\kappa_{\text{wide}}$ and only one iteration) compared to chained NPE in section 5.3.

