# OpenReview forum: "Group equivariant neural posterior estimation"
_ICLR.cc/2022/Conference — ICLR 2022 Poster_

### Official Review · Reviewer_Xr5B · 2021-10-30

**Correctness:** 3
**Technical Novelty And Significance:** 3
**Empirical Novelty And Significance:** 4
**Recommendation:** 5
**Confidence:** 3

**Main Review:**

I found this paper a very enjoyable read, but albeit quite difficult to review for the following reasons. I believe the authors are motivated by extremely interesting problems in physics that leverage non-conventional---atleast in machine learning---datasets like gravitational waves. At best, I can comment from afar that the results look promising and the overall execution is of high quality.

I do have some concerns regarding the technical aspects of the paper though. First, there are a few clarifications that I'd like the authors to address. In all the equivariant literature, including designing equivariant normalizing flows, we are interested in pushing forward a $G$-invariant prior density to a $G$-invariant target. In this paper, the authors instead seek to model a $G$-equivariant density/posterior which is a completely different creature. However, there must be a typo or a misunderstanding on my part as equation 4 is an ***invariant*** distribution and not equivariant as otherwise the group element $g$ must commute and appear in the LHS. This also brings a few points with regards to writing clarity and perhaps mathematical rigor. It is unclear to me what it means for a group element $g$ to act on parameters $\theta$. For the data $x$ this is not an issue as we take a representation of the group and act on the space, but how does one act on the parameters? Do the authors mean acting on the density $p(\theta)$? This leads to my central complaint of the paper which is the use of the word pose. Simply put this is vague in the context of equivariance despite the citation to Jaderberg et. al 2015. It seems like the main thrust of defining pose is so that we can find an inverse group element which can then be used to standardize the pose. By the definition of a mathematical group such an element always exist and as a result I did not follow the discussion in section 3.3 that requires a kernel $\kappa$ over a subset of elements in $G$. Specifically, I do not understand the practical motivation of standardizing the pose to a narrow band.

With regards to the claims about exact and approximate equivariance achieved by GNPE in comparison to prior work I believe the statement is not the best comparison. To be precise, prior work has focused on guaranteeing equivariance by constructing equivariant maps directly while here equivariance relies on finding a pose standardizing element. It is completely unclear how difficult this could be for many groups of interest and as a result I believe we cannot fully generalize this claim, unless the authors provide an efficient algorithm to do so.

I have no specific comments on the experiments but it does seem that GNPE improves over NPE but the data domain is a bit far removed from my area of expertise.



**Summary Of The Paper:**

This paper studies group equivariant neural posterior estimation which seeks to endow conventional NPE method with equivariance of both the data and parameters simultaneously. To test the efficacy of the proposed approach the authors experiment with gravitational wave data and show GNPE achieves considerable performance gains.

**Summary Of The Review:**

Overall the paper is well written and the experiments seem to demonstrate the power of GNPE. There are a few technical questions that the authors could do well to answer which may change my score and opinion of the paper.

---

> ### Author Response · Authors · 2021-11-12
> **Initial response to review**
>
> We appreciate the generally positive appraisal of the paper that it was a "very enjoyable read", motivated by an "extremely interesting" problem. We do apologize for the difficulties encountered in reviewing it, and we do hope that our explanations below (as well as associated changes in the manuscript) will allow the reviewer to evaluate the manuscript more confidently.
>
> “It is unclear to me what it means for a group element $g$ to act on parameters $\\theta$.”
>
> We believe that there are two main points of misunderstanding. The first is how group elements act on parameters: our answer is that we are working with transformation groups that are **defined** by their action on $\theta$, not with abstract groups (e.g., https://en.wikipedia.org/wiki/Group_theory#Transformation_groups). If we were working with abstract groups, then we agree that we would need to specify how $\\theta$ transforms as a representation of G, but in our case it is natural to work with transformation groups so there is already an identification.
>
> To be very precise, consider the harmonic-oscillator example from section 4. Here $\\theta = (\\omega_0, \\beta, \\tau)$. A pose group element $g$ is a time translation $\\Delta \\tau$, which transforms $\\theta$ as $g\\theta = (\\omega_0, \\beta, \\tau + \\Delta\\tau)$.
>
> “Specifically, I do not understand the practical motivation of standardizing the pose to a narrow band.”
>
> The second main question is why we must standardize the pose to within a band rather than exactly. This goes to the heart of the problem, namely that we do not know the pose *a priori*; rather it is to be inferred along with the rest of the parameters. If we knew the pose precisely, then we agree we could standardize it exactly using the inverse group element, however for the problems we are concerned with this is not the case.
>
> As an example, consider the gravitational-waves problem. Here, the pose corresponds to the times of arrival of the signal in each detector. We do not know this information, however, until we analyze the data. But by having this information, we could align the arrival times in each detector at t=0, and thereby simplify the inference task. This is the circular problem that we resolve in this work. (See also our general response to all reviews above.)
>
> “However, there must be a typo or a misunderstanding on my part as equation 4 is an invariant distribution and not equivariant”
>
> We think of the posterior as a mapping from $x$ to (distributions over) $\\theta$, $x\\rightarrow p(\\theta|x)$. This mapping is *equivariant* if $p(\\theta | x) = p(g\\theta|T_g x) |\\det J_g|$; in other words, transforming the data $x$ by $T_g$ can be compensated by transforming the parameters $\\theta$ accordingly. It would be *invariant*, if $p(\\theta | x) = p(\\theta|T_g x)$, i.e., if transforming the data had *no* effect on the posterior. We will aim to clarify this further in the manuscript, but we do think the definition in the paper is both correct and mathematically rigorous.
>
> “With regards to the claims about exact and approximate equivariance achieved by GNPE in comparison to prior work I believe the statement is not the best comparison.”
>
> As stressed in our main reply to all reviewers, there is no equivariant architecture that we are aware of that can enforce such an equivariance for *conditional* normalizing flows. In particular, as stated in our related work section (and highlighted in the review) the equivariant normalizing-flows literature is concerned with a completely different problem.

---

> > ### Comment · Reviewer_Xr5B · 2021-11-21
> > **Response to the rebuttal**
> >
> > I thank the authors for their response and positive engagement overall during the discussions. I believe authors addition of Appendix A.1 and A.2 are a step in the right direction for adding much needed clarity to some of the details from the main paper. Having said that there are still a few remaining points that I am unsure about which currently prevents me from fully endorsing this paper. I will outline them below.
> >
> > - I thank the authors for attempting to clarify how $g$ acts on parameters, but I still don't think this was abundantly clear to me. In the paper the premise relies on both the data and the parameters jointly being equivariant. This would mean that the same group $G$ acts on the data space as well as the parameter space. $G$ could be transformation group (of which matrix groups and permutation groups are an example), but in these cases their action on the underlying object---e.g. set, vector space---is well defined and transformations "preserve this structure". My question still remains what does this mean for the parameter space. In the example of the harmonic oscillator, one could choose to model the time symmetry as a translational symmetry. In this case, I suspect the parameter space can be thought of as again a vector space and then the action of an element is clear. However, this is not stated in general for the parameter space and the reader must make this inference (which might be the wrong inference!).
> > - Overall, I still find the technical presentation of the material to lack sufficient mathematical rigor. This is a point that has been raised by other reviewers and I believe this is the main obstacle in readability. I believe the authors are knowledgeable about the subject but present material in a slightly callous way which prevents a precise understanding of their techniques. Honestly, the notation in this paper is also non-standard and a bit cumbersome. Using $g^{\theta}$ is a bit awkward when what it really seems to mean is that prior to training you do data pre-processing/augmentation for both the parameters. One way to improve readability is to be a bit more precise in the usage of groups (which groups?) and their actions and clearly defining what that looks like. The other issue is that pose standardization is still unclear as it lacks a precise definition. This I believe is the main bottleneck here.
> > - With regards to the authors claims that there exists no equivariant conditional normalizing flow. I didn't fully understand the nuance here:
> > $p(\theta | x) \propto p(x | \theta) * p(\theta)$. To get equivariance, as the authors note we can take an invariant prior over the parameters and an equivariant likelihood. Can we simply not pick an equivariant map to model $p(x| \theta)$ here? What am I missing here? I believe, I have a conceptual issue here and not the authors so I would greatly appreciate the clarification.

---

> > > ### Author Response · Authors · 2021-11-21
> > > **Response to response**
> > >
> > > Thank you for your reply.
> > >
> > > A general remark: GNPE is a technique for **simulation-based inference (sbi)**. In sbi settings, the spaces of parameters $\\theta$ and observations $x$, as well as the forward model mapping $\\theta\\rightarrow x$ are known beforehand. The goal of GNPE is to integrate known domain knowledge (e.g., equivariances).
> > >
> > > 1. *(...) how $g$ acts on parameters, but I still don't think this was abundantly clear to me.*
> > > Possibly a source of confusion is the definition of equivariance we adopt. We consider a forward model $\\theta\\rightarrow x$ equivariant, if a transformation $g$ applied to $\\theta$ transforms a simulation/data $x$ with known transformation $T_g$. Note that $g$ and $T_g$ do not act in the same space: $g$ acts on $\\theta$ and $T_g$ acts on $x$. Sometimes this is referred to as a *covariance* instead (see footnote 1), and equivariance is reserved for situations in which $g=T_g$ (and your comments seem to refer to this alternative definition?).
> > > In sbi settings, the spaces underlying $\\theta$ and $x$ are known beforehand, there is no question about the spaces in which $g$ and $T_g$ act. In the toy example, $g$ indeed acts on the vector space of $\\theta=(\\omega_0,\\beta,\\tau)$ by addition of $\\Delta\\tau$ to $\\tau$, and $T_g$ acts on the time-series $x$ by shifting the data (i.e., relabeling indices) accordingly. This is explicitly shown in equation (10), but we can add a further line to clarify this.
> > > We further believe that your criticism of lack of generality does not apply. GNPE is a technique to integrate *known* equivariances into sbi. We can not provide a general way of coming up with the correct equivariance since this depends on the forward model. In short: yes, the domain expert (or reader, as you put it) applying GNPE to their problem must know the space of $\\theta$ and $x$ (as well as the correct equivariance). But this is true for all sbi methods.
> > > 2. *Overall, I still find the technical presentation of the material to lack sufficient mathematical rigor.*
> > > Our understanding is that the concern of Reviewer W1NJ regarding ‘rigor’ primarily concerned detailed explanations of the equations in section 3, which we addressed by providing explanations and a proof for every equation (see A.1 and A.2).  Could you please be more specific for which statements you have concerns about 'rigor’, or which you regard as 'callous’?
> > > Regarding the notation, it is unclear to us what you mean by 'non-standard’. We denote a group element by $g$ and the representation acting on $x$ by $T_g$, which is standard. Indeed, one can think of applying $-g^\\theta$ and $T_{-g^\\theta}$ in training as a preprocessing operation. Importantly though, this preprocessing operation depends on $\\theta$, and we make this clear with the superscript $\\theta$. What notation would you have found more ‘standard’ for this specific example?
> > > Lastly, we can not provide a more general definition of a pose, as it depends on the forward model and the exact equivariance.  For translational equivariances, the pose would typically be the location of an object (which one would shift to the origin with GNPE). For rotational equivariances, the pose could e.g. be a direction (i.e., an angle). The point is that while the pose is closely related to the underlying equivariance, a meaningful definition depends on the specific inference problem and can only be made with domain knowledge (note that in many problems it will be fine if the defined pose is somewhat arbitrary-- as long as the data can be aligned to some position, it does not matter *which* position it is aligned to). Again, the goal of GNPE is to provide a tool to integrate known domain knowledge; but that domain knowledge is required as input.
> > > 3. *Can we simply not pick an equivariant map to model $p(\\theta|x)$ here?*
> > > For accurate inference, one typically needs powerful density estimators that are expressive enough to model $p(\\theta|x)$, such as normalizing flows which are standard in sbi. We are not aware of a density estimator $q$ that is expressive (e.g., a normalizing flow) and at the same time equivariant (i.e., such that $q(\\theta|x) = q(g\\theta|T_g x) |\\det J_g| ~~\\forall g\in G$ is guaranteed) under the transformation groups we consider in our manuscript. If you are aware of such models, please let us know.
> > >
> > > We believe that your concerns are based on reading the paper from a perspective of general, abstract groups, and for assuming a definition of equivariance that differs from the one used here. However, GNPE is a technique meant for practical applications in sbi settings, in which the domain expert knows the parameter-spaces and sensible group transformations beforehand (as in the problems we used for illustration). We can add a statement to the manuscript to clarify this further.
> > >
> > > Dos this answer your questions?

---

> > > > ### Author Response · Authors · 2021-11-22
> > > > **We will update the presentation of our method**
> > > >
> > > > Dear Reviewer, just a quick update that we are currently in progress of updating sections 3.1-3.3 according to the feedback. We will also add an appendix A.3 where we show that the inferred posterior is indeed equivariant by construction with GNPE (for an exact equivariance).

---

> ### Author Response · Authors · 2021-11-21
> **Kind reminder to update your review**
>
> Dear Reviewer, we provided detailed answers to all of your questions. There have also been several updates in discussion with reviewer tFyT which resulted in them changing their score to 8 ('good paper, accept'), with confidence 4.
>
> We would appreciate a response from you, and in particular whether our response alleviates your concerns and allows you to update your score, or guidance on what additional evidence and clarification you would ask us to provide.

---

> ### Author Response · Authors · 2021-11-22
> **Update of the manuscript**
>
> Dear Reviewer,
>
> We hope that our comments clarified the issues you raised. We reorganized sections 3.2-3.4 and made sure to explicitly emphasize the simulation-based inference context. We also believe these sections are now easier to follow. We now emphasize earlier that we infer the joint posterior $p(\\theta,\\hat g|x)$ over $\\theta$ and $\\hat g$, and show how this helps us to standardize the pose. We also added Fig. 2 to illustrate how GNPE alternately samples $g^\\theta$ and $\\hat g$.
>
> In the new appendix A.3 we now show that the posterior estimated with GNPE can indeed be made equivariant by construction. We hope that these changes address your concern about the rigor of our method.

---

> ### Comment · Area_Chair_nuwc · 2021-11-24
> **Respond to author feedback**
>
> Please respond to author feedback and other reviewers' comments and indicate if it changes your rating.

---

### Official Review · Reviewer_tFyT · 2021-11-02

**Correctness:** 3
**Technical Novelty And Significance:** 3
**Empirical Novelty And Significance:** 4
**Recommendation:** 8
**Confidence:** 4

**Main Review:**

Overall I found the problem and approach to be interesting.  The paper is well-written, but I have a number of comments that I list below.

Major comments:
- Despite being central to the main application and some of the setup of the paper, it was not clear to me what "approximate equivariance" looks like in practice.  For the damped harmonic oscillator toy example, it's clear that the data are exactly jointly equivariant with $\tau$: if $\tau$ is shifted by 1, the data are also shifted by 1.  In the approximately equivariant case, I'm not sure what that would look like.  Is it that the distribution of the other parameters depends on $\tau$ but only weakly?  In the real data example, it's clear what transformation should be applied to the data to put it in the standardized pose for the exact equivariance, but what do the approximately equivariant data transformations look like?
- The "Chained NPE" (equation (12) in Section 5.3) makes sense as a reasonable baseline, but it is unclear to me why it performs so much worse than the wide $\kappa$ GNPE, and it would be good to include more discussion of this point.  In particular, it seems like since the GNPE with wide $\kappa$ only performs one Gibbs sampling iteration, it essentially 1) draws the initial pose estimates from a standard NPE density estimator, 2) draws the a full set of parameters according to equation (8), 3) draws a noisy version of the pose from equation (7), and then 4) redraws the full set of parameters from equation (8).  If I understand the chained NPE approach from its minimal description, it is essentially only performing steps 1) and 2).  If this is the case, then it feels like having a wide $\kappa$ is just "blurring" the estimate of the pose relative to the chained NPE approach, in which case the increased performance going from chained NPE to wide $\kappa$ NPE is attributable just to having a more diffuse estimate of the pose in the latter.  Does this indicated that chained NPE is overly confident in its estimation of the pose?  Could chained NPE be "fixed" by learning a more expressive posterior distribution (e.g., mixture of Gaussians instead of a single Gaussian)?  I ask because chained NPE is conceptually much simpler and doesn't require the additional tuning parameter of $\kappa$, or Gibbs sampling, making chained NPE -- at least in theory -- much more attractive.

Minor Comments:
- Beyond the standard issues with choosing neural network architecture, this approach introduces two new algorithmic parameters that need to be tuned: 1) the "noise tolerance" distribution in the pose estimation, $\kappa(\epsilon)$ and 2) the number of Gibbs Sampling steps that are needed to reach convergence.  2) does not seem so problematic in practice (there are various methods of diagnosing the convergence of Markov chains) although it would be good to describe how this was chosen for this paper (e.g., on p. 8 it is stated that "We find that fast-mode GNPE converges in one iteration, whereas accurate-model requires 30." but it is not stated what was used to determine this).  To me 1) seems more problematic since it is (in general) a distribution, possibly over multiple dimensions.  Is it sufficient to only consider uniform distributions, as in the present paper, or would other distributions result in better performance?  How should one set the parameters of such distributions?
- In the paragraph following equation (8) on p. 5, burn-in and thinning only result in approximately independent samples.  Similarly, It should be noted that equations (7) and (8) only result in approximate samples from the posterior (due to 1) the asymptotics of Gibbs sampling; and 2) any potential mismatch between the variational approximation $q(\theta | x, \hat{g})$ and the true posterior $p(\theta | x, \hat{g})$ due to e.g., the variational family not being rich enough, an insufficiently flexible inference network that cannot learn the amortized posterior, not enough samples when training the inference network, or failure in the optimization of the neural network).
- I find the c2st metric a bit difficult to interpret.  It would also be good to consider more interpretable metrics, like the MSE of the posterior mean for each method, as well as e.g., the coverage of credible intervals.
- In contrast, Figure 4 is very nice and easy to interpret.  It would be helpful to also visualize these same marginal posteriors for the "wide $\kappa$" and "NPE (chained)" methods to see the effect of $\kappa$ (see above comment).


Typo:
- on p. 9 "GNPE achieves a scores" --> "GNPE achieves a score"

**Summary Of The Paper:**

In this paper the authors present a method for performing Bayesian inference in likehood-free settings where the data and parameters are jointly equivariant or approximately equivariant.  Examples include translational or rotational equivariance, where if the latent parameter are translate or rotated, then the distribution over the data is the same, except that the data are translated or rotated in the same way as the latent parameters.  The method presented here acts by mapping each simulation to a standard "pose" and then aims to learn both the original pose and any additional parameters from the data.  There are two main components to this method -- 1) the standard simulation-based inference method of estimating latent parameters from data, but applied to data that have had their pose standardized and 2) a Gibbs sampling framework to use the family of posteriors learned from the first component to jointly estimate the pose and ant other parameters from unposed data.

**Summary Of The Review:**

The paper is well-written and presents an interesting method for a methodologically interesting problem with a nice application.  Some of the key aspects of the paper (e.g., what does an approximate equivariance look like in practice? why does the method outperform chained NPE? how should one set the kernel width?) are a bit unclear, but overall the paper is a good contribution.

---

> ### Author Response · Authors · 2021-11-12
> **Initial response to major comments**
>
> We appreciate the generally positive appraisal of the paper, that the approach is “interesting” and that the paper is “well written”. We aim to clarify the specific questions here, and will update the manuscript accordingly. In particular we provide additional plots for major comment 2, and minor comments 3 and 4.
>
> Major comments:
>
> 1. “it was not clear to me what "approximate equivariance" looks like in practice.”
> We agree that the paper would benefit from a precise definition of “approximate equivariance”. Strictly speaking, this refers to any joint transformation of $x$ and $\\theta$ that is not an exact equivariance. However, for this to be useful in practice, an approximate equivariance must *simplify the data.* For gravitational waves, the sky rotation / relative time shift is an approximate equivariance: the predominant effect of changing the sky position is to shift the signal arrival times in the individual detectors; however, this also subdominantly affects the linear combination of polarization modes measured (i.e. there is an additional, but more subtle effect, as the measured waveforms are also modulated by the angle of incidence). Aligning the arrival times nevertheless corresponds to an important simplification of the data representation.
> For the harmonic oscillator, the exact time-translation equivariance could be made approximate by including, e.g., $\\tau$-dependent noise. Nevertheless, the excitation time $\\tau$ could still be set to 0 to align the data. It would be much easier for an inference network to process the (approximately) aligned data than unaligned data.
>
> 2. “The "Chained NPE" (equation (12) in Section 5.3) makes sense as a reasonable baseline, but it is unclear to me why it performs so much worse than the wide κ GNPE, and it would be good to include more discussion of this point.”
> We agree that chained NPE seems appealing, and in fact initially tried this approach in the early stages of this project. The main density estimator is trained with data with an entirely standardized pose, so it is very powerful. We found that this method however has one major flaw: Inaccuracies in the initial pose estimate propagate through to inaccuracies in the other parameters [as the reviewer correctly pointed out, see also equation (12)]. In practice, if standard NPE is not sufficiently accurate for the full parameter space, it is likely also inaccurate when only estimating the pose. We are demonstrating this empirically in this plot https://tinyurl.com/y5h8a6xz, which we add to the appendix of the manuscript. We display chained NPE results for the four parameters from Figure 4, and additionally show the pose $(t_H, t_L)$. We observe that the pose inferred with $q_{init}$ using NPE (green) differs from the true pose (teal). This leads to an inaccurate estimate of the remaining parameters. When using the “oracle’’ pose, the same density estimator provides much better results.
> This problem could not easily be solved by using a better density estimator to infer the pose--in fact, we already use a very expressive neural spline flow for the initial pose estimate, and the pose estimate displayed in the plot is the best we achieved with extensive optimisations.
> GNPE with one iteration, as the reviewer pointed out, works much better. It can be understood as a more robust version of chained NPE; the only difference is that GNPE is conditioned on the pose *proxy* instead of the pose itself. Intuitively, the blurring operation to obtain the pose from the proxy helps against the overconfidence of the network in a potentially inaccurate pose estimate. More rigorously, denoting the NPE pose estimate with $q_{init}$, the true pose distribution with $p$, a standard convolution with $\\ast$ and the KL divergence with $KL$, the data processing inequality implies $KL(q_{init}\\ast\\kappa||p\\ast\\kappa)\\leq KL(q_{init}||p)$. In other words, the NPE proxy estimates the true proxy better than (or at least as good as) the NPE pose estimates the true pose (in terms of $KL$). We illustrate this with this plot https://tinyurl.com/56zv2uec, which we add to the appendix of the manuscript.

---

> > ### Comment · Reviewer_tFyT · 2021-11-19
> > **Response to response**
> >
> > Thank you for your thorough response to my major and minor comments.
> >
> > I am satisfied with all of the responses to the minor comments.  I still find the performance gains discussed in response to my major comment 2 a bit mysterious, but thank you for the response.

---

> > > ### Author Response · Authors · 2021-11-20
> > > **Response to response to response**
> > >
> > > Dear Reviewer, thank you for your response. Would you be able to point us to which aspect of our response to comment 2 you find 'mysterious', so that we can attempt to explain it more clearly? More generally, are there any further explanations or analyses that you would ask us to do in order for you to to be able to adjust your score?

---

> > > > ### Comment · Reviewer_tFyT · 2021-11-20
> > > > **Response to response to response to response**
> > > >
> > > > I apologize for the unclear response.  I don't find your response mysterious, but the phenomenon that GNPE with one Gibbs sampling step seems to outperform chained NPE in your experiments is still counterintuitive to me.
> > > >
> > > > One thing that I am hung up on is that since the neural estimate of the posterior over the pose -- in the toy model $q(\tau | x)$ -- is only used to initialize the pose in the Gibbs sampling, it seems like the stationary distribution of the Gibbs sampler should be independent of the choice of $q(\tau | x)$.  This means that all of the information about the pose must be coming from $q(\theta' | x')$, but the pose does not appear there.  So it feels like for the toy data this approach should not be able to uncover the pose, and if it can recover the pose then that just means that the Gibbs sampler has not converged.  Am I missing something obvious?
> > > >
> > > > One more very minor, unrelated point -- I should also say that I'm not sure what you mean in the paper or in your "Response to response" when you say "this also subdominantly affects the linear combination of polarization modes measured".  What does "subdominant" mean in this context?

---

> > > > > ### Author Response · Authors · 2021-11-20
> > > > > **Chained NPE vs GNPE**
> > > > >
> > > > > Thanks for the reply. I think that understanding why GNPE performs better than chained NPE gets to the heart of why our method works so well. Essentially, it is easier to estimate the pose itself if the data have been approximately aligned. Chained NPE estimates the pose using non-aligned data, whereas GNPE self-consistently aligns the data while estimating the pose. This is the chicken-and-egg problem that GNPE resolves using Gibbs sampling.
> > > > >
> > > > > For the harmonic oscillator, you are absolutely correct that the stationary distribution of the Gibbs sampler is independent of $q(\\tau | x)$: the sole purpose of this estimate is to initialize the Gibbs sampler so that fewer iterations are required for convergence. All information about the pose therefore comes from $q(\\theta'|x')$ *and* $\\kappa(\\epsilon)$, which, together, give the joint posterior over $\\theta$ and $\\hat\\tau$. We sample this using Gibbs. The pose $\\tau$ is explicitly calculated in step 3 on page 6, when we "undo the time translation": we know $\\hat\\tau$ (from step 2), and we sample $(\\tau - \\hat\\tau)$ from $q(\\theta'|x')$, so adding these we get the pose $\\tau$. Intuitively, we know how much we time-shifted the data to make this estimate of the residual, so we can just undo this shift in the parameter in the end.
> > > > >
> > > > > Does this make sense?
> > > > >
> > > > > Regarding your other question, gravitational waves include two polarization modes. A detector measures a linear combination of these, depending on the angle of impingement. A small change in sky position therefore causes a change in this linear combination, which results in a small change to the waveform measured. This is generally less important than the dominant effect of changing the time of arrival of the signal.

---

> > > > > > ### Comment · Reviewer_tFyT · 2021-11-20
> > > > > > **Reply to Chained NPE vs GNPE**
> > > > > >
> > > > > > I think I remain unclear on how GNPE can learn the pose in the toy model.
> > > > > >
> > > > > > Let me try again -- Here is a simplified model that captures my unease (or misunderstanding):
> > > > > >
> > > > > > $\tau \sim \mathcal{N}(-5, 1)$
> > > > > >
> > > > > > $X | \tau \sim \mathcal{N}(\tau, 1)$
> > > > > >
> > > > > > This is equivariant with respect to translation in that the distribution of $X | \tau$ shifts by $c$ if $\tau$ shifts by $c$.
> > > > > >
> > > > > > Now, let's take $\epsilon \sim \mathcal{N}(0, 1)$, and as in the toy model write $\hat{\tau} = \tau + \epsilon$.  Write $\tau' = \tau - \hat{\tau} = -\epsilon$ and $X' = X - \hat{\tau}$
> > > > > >
> > > > > > If I train an NPE on pose-shifted data to learn $q(\tau' | X')$ I can see that since
> > > > > >
> > > > > > $\tau' \sim \mathcal{N}(0, 1)$
> > > > > >
> > > > > > $X' | \tau' \sim \mathcal{N}(\tau', 1)$
> > > > > >
> > > > > > we get that $\tau' | X' \sim \mathcal{N}(X' / 2, 1/2)$, so we'll assume that our NPE estimate learns that true posterior.  Now I'll follow the steps on page 6:
> > > > > >
> > > > > > 1.  We agree that the initialization doesn't matter if I run my Gibbs sampler long enough, so let's take $\tau^{(1)} = 0$.
> > > > > >
> > > > > > 2.  I sample $\epsilon^{(1)} \sim \mathcal{N}(0, 1)$ and set $\hat{\tau}^{(1)} = \tau^{(1)} + \epsilon^{(1)}$, and $X'^{(1)} = X - \hat{\tau}^{(1)}$.
> > > > > >
> > > > > > 3.  I sample $\tau'^{(1)} \sim \mathcal{N}(X'^{(1)}/2, 1/2)$, and set $\tau^{(2)} = \tau'^{(1)} + \hat{\tau}^{(1)}$.
> > > > > >
> > > > > > Then I repeat steps 2 and 3.  I implemented this in python (code below), and the stationary distribution of the Gibbs sampler over $\tau$ converges to $\mathcal{N}(X, 1)$.  Note that this is _not_ the true posterior, which is
> > > > > > $\mathcal{N}((X-5)/2, 1/2)$.  And the blurring distribution on $\epsilon$ contributes to the posterior variance being inflated, but not to the fact that the posterior mean is wrong.  It makes sense to me that the true posterior on the pose can't really be obtained here because $-5$ -- the prior mean on $\tau$ -- doesn't appear anywhere in steps 1, 2, or 3.
> > > > > >
> > > > > >
> > > > > > ```
> > > > > > import numpy as np
> > > > > > import scipy.stats
> > > > > >
> > > > > >
> > > > > > np.random.seed(42)
> > > > > > x = 3
> > > > > > tau = [0]   # Step 1
> > > > > >
> > > > > > for i in range(1000000):
> > > > > >     # Step 2
> > > > > >     epsilon = np.random.normal(loc=0, scale=1)
> > > > > >     tau_hat = tau[-1] + epsilon
> > > > > >     x_prime = x - tau_hat
> > > > > >     # Step 3
> > > > > >     tau.append(np.random.normal(loc=x_prime/2,
> > > > > >                                 scale=np.sqrt(0.5))+tau_hat)
> > > > > >
> > > > > > # Check if this CDF (after burn-in) is close to N(x, 1)
> > > > > > def my_cdf(val):
> > > > > >     return scipy.stats.norm.cdf(val, loc=x, scale=1)
> > > > > > print(scipy.stats.kstest(np.array(tau[1000::100]), my_cdf))
> > > > > > >>> KstestResult(statistic=0.00529800193900376, pvalue=0.9404140198897314)
> > > > > > ```

---

> > > > > > > ### Author Response · Authors · 2021-11-20
> > > > > > > **GNPE works on toy example**
> > > > > > >
> > > > > > > Dear Reviewer,
> > > > > > >
> > > > > > > Thank you for the interesting discussion, and for taking the effort to come up with this clarifying toy example!
> > > > > > >
> > > > > > > We can confirm that GNPE works for this toy example (see code at https://tinyurl.com/3z2j7w63), but one needs to use the equivariance relationship of the posterior - not that of the likelihood - for GNPE:
> > > > > > >
> > > > > > > In your toy example, the forward model is equivariant under
> > > > > > > $\\tau\\rightarrow\\tau+\\Delta\\tau,~x\\rightarrow x+\\Delta\\tau$.
> > > > > > > If the prior was invariant, then this equivariance would indeed be inherited by the posterior (as we show in the new appendix A.1). However, the prior is not invariant.
> > > > > > >
> > > > > > > Interestingly though, the posterior still has an equivariance, but a different one. It is equivariant under
> > > > > > > $\\tau\\rightarrow\\tau+\\Delta\\tau,~x\\rightarrow x+2\\Delta\\tau$,
> > > > > > > so $x$ transforms under a *different representation*. As stated around equation (4) in the manuscript, the equivariance of the posterior is the relevant one for GNPE, so this is the one we need to use.
> > > > > > >
> > > > > > > In order to fix your GNPE implementation, we thus only require to change the line `x_prime = x - tau_hat` to `x_prime = x - 2 * tau_hat`.
> > > > > > > There is one more little mistake in your experiment: the posterior $p(\\tau’|x’)$ has the same form as $p(\\tau|x)$ due to the equivariance relation, i.e., $p(\tau’|x’) = \\mathcal{N}((x-5)/2,1/2)$, you forgot the $-5$. To fix that, we need to change `np.random.normal(loc=x_prime/2` to `np.random.normal(loc=(x_prime - 5)/2`.
> > > > > > >
> > > > > > > With these fixes, GNPE infers the correct posterior. We will add this as an instructive example to the appendix of the manuscript. For a draft including a few further explanations and visual confirmation that GNPE infers the correct posterior see here: https://tinyurl.com/yp9m2cpd.
> > > > > > >
> > > > > > > Does this answer your questions?
> > > > > > >
> > > > > > > ```
> > > > > > > import numpy as np
> > > > > > > import scipy.stats
> > > > > > >
> > > > > > >
> > > > > > > np.random.seed(42)
> > > > > > > x = 3
> > > > > > > tau = [0]   # Step 1
> > > > > > >
> > > > > > > for i in range(1000000):
> > > > > > >     # Step 2
> > > > > > >     epsilon = np.random.normal(loc=0, scale=1)
> > > > > > >     tau_hat = tau[-1] + epsilon
> > > > > > >     # x_prime = x - tau_hat
> > > > > > >     x_prime = x - 2 * tau_hat
> > > > > > >     # Step 3
> > > > > > >     # tau.append(np.random.normal(loc=x_prime/2,
> > > > > > >     # 						      scale=np.sqrt(0.5))+tau_hat)
> > > > > > >     tau.append(np.random.normal(loc=(x_prime - 5)/2,
> > > > > > >     							scale=np.sqrt(0.5))+tau_hat)
> > > > > > >
> > > > > > > tau_samples = np.array(tau[1000::10])
> > > > > > > print(f'samples mean: \t{np.mean(tau_samples):.3f}')
> > > > > > > print(f'samples std: \t{np.std(tau_samples):.3f}')
> > > > > > > >>>samples mean: 	-0.997
> > > > > > > >>>samples std: 	 0.707
> > > > > > >
> > > > > > > # Check if this CDF (after burn-in) is close to N(x, 1)
> > > > > > > def my_cdf(val):
> > > > > > >     return scipy.stats.norm.cdf(val, loc=x, scale=1)
> > > > > > > print(scipy.stats.kstest(np.array(tau_samples), my_cdf))
> > > > > > > >>> KstestResult(statistic=0.9812743278989882, pvalue=0.0)
> > > > > > > ```

---

> > > > > > > > ### Comment · Reviewer_tFyT · 2021-11-20
> > > > > > > > **Thank you**
> > > > > > > >
> > > > > > > > I see.  Thank you for taking the time to respond.
> > > > > > > >
> > > > > > > > I now appreciate that the relevant equivariance is that of the posterior, not of the likelihood (which does contrast to some extent with the exposition of the paper), although I do agree that an equivariant posterior would fall out for a model with an equivariant likelihood with a prior that is uniform over pose.  I find this latter use-case far more compelling anyway -- if the pose is a nuisance parameter it makes sense to place a flat prior on it, and we often understand the forward model well enough to know if it is equivariant or not.
> > > > > > > >
> > > > > > > > Thank you again for engaging, and taking the time to respond.  I've raised my score to "accept"

---

> > > > > > > > > ### Author Response · Authors · 2021-11-21
> > > > > > > > > **Thank you!**
> > > > > > > > >
> > > > > > > > > Dear Reviewer, thank you for engaging. We appreciate the effort you put into contributing to the discussion.

---

> ### Author Response · Authors · 2021-11-12
> **Initial response to minor comments**
>
> Minor comments:
>
> 1. “[Regarding the kernel:] How should one set the parameters of such distributions?”
> The kernel should always be symmetric and centered around the neutral group element. Beyond that, we believe that it is best to keep the kernel as simple as possible, in particular we advise to use a factorized kernel for multiple dimensions (even though more complicated choices would in principle be possible). The simpler the kernel, the better one can interpret the results. This leaves us with two natural choices for the kernel $\\kappa$: a uniform or a Gaussian distribution. The earlier has the advantage of a better worst-case scenario (in our application the flow thus knows that the time shift of the input GW will never be more than 1ms). In our toy example on the other hand, we use a Gaussian kernel in order to preserve the Gaussian nature of the posteriors. In practice, we did not find a significant difference in performance.
> The kernel width is an additional hyperparameter which, as described in section 3.4 paragraph 3, should be chosen to accommodate the practical trade-off between convergence speed and accuracy: In time critical applications, the kernel should be chosen wide enough such that convergence is typically achieved after one GNPE iteration  (which can be checked without ground-truth posteriors by evaluating the convergence behaviour on a simulation). On the other hand, choosing a smaller kernel will improve performance (as shown in our results in section 5) but inference will take longer. While the optimal choice of the kernel depends on the specific application, as a rule of thumb, the kernel width should be on the same scale as the typical standard deviation of the pose posterior. We add a comment with guidance on how to choose the kernel, and in particular its width, to the manuscript.
> “[Regarding number of GNPE iterations and convergence criteria:] it would be good to describe how this was chosen for this paper”
> One can indeed assess the convergence quantitatively--e.g., by comparing the JS divergences between the inferred pose posteriors between two successive GNPE iterations. In practice we used such criteria to get a feeling of the required number of iterations. However, in the end we fixed the number of iterations to 30 to provide a run time guarantee to our experimental colleagues. We will add a remark clarifying this to the manuscript.
>
> 2. “burn-in and thinning only result in approximately independent samples. (...) (7) and (8) only result in approximate samples from the posterior”
> Yes, we will add a brief statement to clarify this.
>
> 3. “I find the c2st metric a bit difficult to interpret.”
> C2st is a very sensitive measure of deviation, which can make *high* c2st scores hard to interpret -- e.g. a c2st score of 1 (‘perfect discriminability’) can already occur if just one out of many dimensions is poorly estimated, even if all the others match perfectly.  However, a *low* c2st score implies that the distributions match very well, and that there is no deviation that could be exploited by a neural network to separate them (which typically also means that they are visually identical).
> C2st scores on the level that we report are *extremely* hard to obtain in simulation-based inference (even in much simpler applications, see e.g. Lueckmann et al., 2021), since they are sensitive to every parameter and even high-dimensional correlations. We thus want to keep this result in the main text, also to facilitate comparisons with other simulation-based inference (sbi) studies.
> However, we agree that a metric that is better known outside of the sbi community should also be reported. We will add results for the MSE between the (normalized) means of the reference samples and the inferred samples to the appendix (see https://tinyurl.com/4xr592k2) which lead to qualitatively similar conclusions.
>
> 4. “In contrast, Figure 4 is very nice and easy to interpret. It would be helpful to also visualize these same marginal posteriors for the "wide κ" and "NPE (chained)" methods to see the effect of κ”
> Thanks for the suggestion. We will add this figure to the appendix (see https://tinyurl.com/rp8m39za), and indeed it shows that chained NPE deviates substantially from MCMC.

---

### Official Review · Reviewer_NKhw · 2021-11-02

**Correctness:** 3
**Technical Novelty And Significance:** 3
**Empirical Novelty And Significance:** 3
**Recommendation:** 5
**Confidence:** 2

**Main Review:**

Strengths:
- The approach is independent of neural architectures and does not necessitate knowledge of exact equivariances.
- The method seems to be much better than regular NPE in cases where there are known equivariances.

Questions:
- I found the writing in Sections 3.4 and 4 quite confusing to follow. I am not sure how changing $g_\theta$ to the approximate $\hat{g}$ solves the problem of not having "pose" data in images.
- How is $q_{init}$ trained? Is this a regular NPE trained on the same data used for training invariances?
- I think Equation 9 needs further discussion in the main body. What is $Q$ and how is it related to lowercase $q$?
- I think it would be really helpful to show in some toy cases how $q(\theta | x , \tau)$ depends on $\tau$ (or $\hat{g}$).

**Summary Of The Paper:**

The paper proposes to incorporate invariances of posterior estimates via a factored posterior estimation approach rather than the standard architectural interventions. The paper proposes to learn an approximate standardizing transformation over data and then learning posteriors over the approximate standardized data.

**Summary Of The Review:**

I am awaiting author responses at this point and will update my review based on their explanations.

---

> ### Author Response · Authors · 2021-11-12
> **Initial response to review**
>
> Responses to specific questions:
>
> 1. “I found the writing in Sections 3.4 and 4 quite confusing to follow. I am not sure how changing gθ to the approximate g^ solves the problem of not having "pose" data in images.”
> We agree that some of this text could be improved, and we  are already working on improving it. This question is central to our method, and should be made clear. As stated above, our goal is to **simultaneously infer the pose of a signal and to use that inferred pose to standardize (or align) the data so as to simplify the analysis**. Our method tackles this by learning a posterior $p(\\theta, \\hat g | x)$ over an expanded parameter space that is augmented by an additional parameter, namely  the approximate pose $\\hat g$.  This joint posterior arises by combining the definition of $\\hat g$ with the neural density estimator $q$, and can be sampled using Gibbs sampling [eq. (7)-(8)]. It is precisely this Gibbs sampling that solves the “chicken and egg” problem of needing an (approximate) pose to align the data and aligning the data with an (approximate) pose.
> The “trick” is that since $q$ is conditioned on $\\hat g$ we are allowed to apply a transformation that depends on $\\hat g$ to the data. Using $g^\\theta$ instead of $\hat g$ to transform the data is equivalent to taking the blurring kernel $\\kappa$ to be a delta distribution, which, as noted in the text, would result in a Markov chain that remains stuck at its starting position in “pose space”. Intuitively, the width of the kernel corresponds to how far the estimate of the pose can correct itself in each Gibbs iteration. In summary, by replacing the pose with the pose proxy, we enable an iterative interplay between the fixed kernel $\\kappa$ and the density estimator $q$ (trained on pose standardized data), which allows for simultaneous standardization of the pose and inference of $\\theta$ (including also the remaining non-pose parameters).
>
>
> 2. “How is qinit trained? Is this a regular NPE (...)?”
> As pointed out in section 3.4 (page 5, above equation 9) and 5.2 (page 8, second paragraph), $q_{init}$ is trained using standard NPE. We add an additional remark in section 4.
>
> 3. “What is Q and how is it related to lowercase q?”
> Equations (7) and (8) describe how we create a Markov chain using Gibbs sampling. As stated above (9), we parallelize sampling by constructing N Markov chains. $Q_j(\\theta|x)$ is the distribution over $\\theta$ represented by taking the $j$th element of each Markov chain. In iteration 0, we initialize the pose estimates via sampling $g^\\theta\\sim q_{init}(g^\\theta|x)$. We will slightly modify the paragraph around (9) to make this clearer.
>
> 4. “I think it would be really helpful to show in some toy cases how q(θ|x,τ) depends on τ (or g^).”
> We will add a figure visualizing such correlations for the GW application to the appendix, see https://tinyurl.com/3ncuhnsz. (We do this for the real example since the toy example has uncorrelated parameters.) This figure is an extension of Fig. 4; in addition to four non-pose parameters it displays the pose proxy $\hat{t}_H$ and the inferred pose $t_H$. This plot shows a strong correlation between $\hat{t}_H$ and $t_H$, which is expected as $\hat{t}_H$ is defined via the kernel as a blurred version of the true pose $t_H$. For the same reason, the observed correlation between $\hat{t}_H$ and non-pose parameters is similar to the correlation between $t_H$ and non-pose parameters. We hope this provides helpful intuition.

---

> ### Author Response · Authors · 2021-11-21
> **Kind reminder to update your review**
>
> Dear Reviewer, we provided detailed answers to all your questions 1-4. There have also been several updates in discussion with reviewer tFyT which resulted in them changing their score to 8 ('good paper, accept'), with confidence 4.
>
> You mentioned you are “awaiting author responses” to update your review. We would appreciate a response from you, and in particular whether our response alleviates your concerns and allows you to update your score, or guidance on what additional evidence and clarification you would ask us to provide.

---

> ### Author Response · Authors · 2021-11-22
> **Update of the manuscript**
>
> Dear Reviewer,
>
> we would like to point you to the changes made to the manuscript to address your concerns. We hope that you are satisfied with our explanations and that this allows you to update your review accordingly.

---

> ### Comment · Area_Chair_nuwc · 2021-11-24
> **Respond to author feedback**
>
> Please respond to author feedback and other reviewers' comments and indicate if it changes your rating.

---

### Official Review · Reviewer_W1NJ · 2021-11-04

**Correctness:** 3
**Technical Novelty And Significance:** 3
**Empirical Novelty And Significance:** 3
**Recommendation:** 6
**Confidence:** 2

**Main Review:**

The major disadvantage of the paper is the writing, which makes the paper difficult to read and evaluate. I can understand the key idea of the GNPE, and the problem investigated by this paper is certainly important. But I cannot check the correctness of the conclusions and algorithm due to lack of necessary assumptions and derivation steps.

Some comments:

1) Where does Eq. (4) come from? I think this is a corollary of several assumptions including the equivariance of prior distribution of theta. If so, please provide all the related assumptions and the proof, which are not trivial for readers.

2) It is unclear why we need \epsilon and \hat g. Are they introduced for the case of approximate equivariance. The first two paragraph in Section 3.3 seems to discuss the necessity of introducing them, but it is really hard to understand. Is it possible to make the explanation more clear by using the toy example?

3) What is the definition of p(\theta|x,\hat g) in Eq. (5)? How is Eq. (5) proved? In addition, please provide proofs of all equations in Section 3.

4) Is there any way for users to choose the distribution of \epsilon in applications?

5) In experiments, only NPE and GNPE are considered. It is worth comparing the performance of GNPE versus other flow models with equivariant architecture.

**Summary Of The Paper:**

The authors propose group equivariant neural posterior estimation (GNPE), a posterior estimation method which can self-consistently infer parameters and standardize the pose. For equivariant posterior distributions, the GNPE can achieve better performance than the traditional NPE. Moreover, GNPE can also be applied to cases where the equivariance of the posterior is approximately estimated.

The major advantage of GNPE over the other flow based models under equivariance constraints is that GNPE is architecture-indepenent, i.e., an arbitrary flow model can be utilized in GNPE and we don't need to design a special network structure for the equivariance.

**Summary Of The Review:**

It is an interesting paper, and possibly make important contributions. For now, I give the score "marginally below the acceptance threshold" due to the poor writing. But if the authors can well explain all the related conclusions and definitions, I can consider to increase the score.

After reading the rebuttal and explanations from authors, I think the revised manuscript is much more understandable, and I change the score to "marginally above the acceptance threshold".

---

> ### Author Response · Authors · 2021-11-12
> **Initial response to review**
>
> We regret that the reviewer had a difficult time reading and evaluating the paper. We hope they will find our additional explanations and associated updates to the manuscript convincing. In particular, we will add two new appendices (A.1 and A.2, see https://tinyurl.com/42rvjp96) in response to requests for further derivations.
>
>
> 1. “Where does Eq. (4) come from?”
> This is the definition of a posterior equivariant under joint transformations of data $x$ and parameters $\\theta$. It arises from the change-of-variables rule for probability distributions under this transformation. We will modify the text to clarify this.
> As noted in footnote 3, one is typically concerned with equivariant *likelihoods*, but this equivariance is inherited by the posterior if the prior is invariant. We add a proof of this in A.1 (https://tinyurl.com/42rvjp96).
>
> 2. “Why do we need $\\epsilon$ and $\\hat g$?”
> We agree that the start of Sec. 3.3 should be improved, and will re-work it. The parameters $\\hat g$ and $\\epsilon$ (which is used to define $\\hat g$) are needed to describe a “blurred” approximate pose. This is a key aspect of our construction, for both exact and approximate equivariances. We have no access *a priori* to the *exact* pose $g$ so we standardize our data with respect to the *approximate* pose $\\hat{g}$. We then (1) infer $\\theta$ based on data standardized with $\\hat{g}$ and (2) infer an updated $\\hat{g}$ from $\\theta$. We iterate (1) and (2) until convergence. In other words, by combining our definition of $\\hat g$ with a learned posterior conditioned on $\\hat g$ we obtain a joint posterior over $\\theta$ and $\\hat g$, which we can sample from using Gibbs sampling.
>
>
> 3. “What is definition of $p(\\theta|x,\\hat g)$ in Eq. (5)?”
> As stated above, $p(\\theta|x,\\hat{g})$ is the posterior $p(\\theta|x)$ which is also conditioned on $\\hat g$, i.e., it is the probability of $\\theta$ given $x$ and $\\hat g$ (which itself is defined in the previous paragraph). This can be related to $p(\\theta|x)$ and $\\kappa(\\epsilon)$. We show this in the new appendix A.2 (https://tinyurl.com/42rvjp96), where we also prove that $p(\\theta|x,\\hat{g})$ is indeed equivariant.
> “Provide proofs of all equations in Section 3.”
> - Eq. (1) is  from (Papamakarios & Murray, 2016) as cited in the submission.
> - (2) and (3) are definitions.
> - We add derivations for (4) and (5) in A.1 and A.2 (https://tinyurl.com/42rvjp96), respectively.
> - (6) is a modification of the NPE loss (1) to learn our defined posterior: we optimize the log-loss of the $\\hat{g}$-transformed parameters given the $\\hat{g}$-standardized data, and additionally condition on the pose proxy $\\hat{g}$ to allow for approximate equivariances. The left column in (7) and (8) is the standard procedure for Gibbs sampling (as explained, e.g., in Gelman 2013). The right column contains a constructive definition for sampling the corresponding random variables, and is a direct consequence of the definition of $\\hat{g}$ in Sec. 3.3. Eq. (8) further assumes that the trained $q$ approximates $p$ well, and we add a corresponding comment to the manuscript.
> - As stated in the manuscript, (9) is derived in A.1 (now A.3).
>
>
> 4. “How to choose distribution of $\\epsilon$?”
> We recommend zero-centered uniform or Gaussian kernels. The width is a hyperparameter which, as described in Sec. 3.4 Para 3, should be chosen to trade-off convergence speed and accuracy. In time critical applications, the kernel should be wide enough such that convergence is typically achieved after one GNPE iteration. On the other hand, a smaller kernel will improve performance (as shown in Sec. 5) but inference will take longer. While optimal choice of the kernel depends on the application, as a rule of thumb, the width should be on the same scale as the typical standard deviation of the pose posterior. We will add guidance on how to choose the kernel width.
>
>
> 5. “Compare performance of GNPE to other flows with equivariant architecture.”
> We are not aware of flows that incorporate equivariances between data $x$ (i.e., the **conditional** variable) and parameters $\\theta$. (See comments to Xr5B and general comments: we are concerned with equivariant **conditional** distributions.) The closest that can be achieved is to use an equivariant embedding network for the flow, however there is no general way (except for GNPE) to enforce the equivariance  *through* the flow. Nevertheless, an equivariant embedding can simplify the task for the flow to learn the equivariance: On the toy model (Sec. 4), a convolutional embedding network (NPE-CNN in Fig. 2) achieves similar performance to GNPE. For GWs (Sec. 5), we are not aware of an embedding architecture equivariant under the local phase shifts described in 5.1. Furthermore, this equivariance is not exact, so an architecture that enforces it would introduce errors. GNPE, by contrast, allows for equivariance breaking effects by conditioning on the proxy.

---

> > ### Comment · Reviewer_W1NJ · 2021-11-22
> > **response**
> >
> > I agree with Reviewer Xr5B that the revision improve the comprehensibility of the paper, although it is still difficult for me to understand. In my opinion, more clear mathematical notation could be helpful for this paper. In addition, I cannot understand why \hat g is required even for the exact equivariant problem. I will spend more time reading this paper.

---

> > > ### Author Response · Authors · 2021-11-22
> > > **We will update section 3 with further explanations**
> > >
> > > Dear Reviewer, thank you for your reply. The problem is that $\\theta$ and thus $g^\\theta$ is unknown at inference time. While we could easily apply $(g^\\theta)^{-1}$ and the corresponding representation $T_{(g^\\theta)^{-1}}$ at train time (when ground-truth $\\theta$ is available for a simulation $x$), we could not use the trained network at inference time (where $\\theta$ is unknown), since then we do not know $g^\\theta$.
> > >
> > > Our resolution is the introduction of $\\hat g$ and Gibbs sampling. By defining $\\hat g$ as a blurred version of $g^\\theta$ (with blurring kernel $\\kappa$), we enable an iterative interplay between $g^\\theta$ and $\\hat g$. Given $g^\\theta$, we can infer $\\hat g$ (by its definition); given $\\hat g$ we infer $g^\\theta$ (or in practice $\\theta$) with a neural density estimator $q(g^\\theta|x,\\hat g)$. The key is that $q$ is conditioned on $\\hat g$, which allows us to (approximately) standardize the pose by applying $T_{\\hat g^{-1}}$ to the data $x$. In other words, $\\hat g$ enables the preprocessing operation with $T_{\\hat g^{-1}}\\approx T_{(g^\\theta)^{-1}}$ despite unknown $g^\\theta$.
> > > The tighter the kernel $\\kappa$, the closer is $\\hat g$ to $g^\\theta$, but this comes at the cost of a higher convergence time of the Gibbs sampler (see new Figure 2, soon to come). In the limit of a $\\delta$ distribution for $\\kappa$, we have $\\hat g=g^\\theta$, but the chain does not deviate from its original position anymore, and GNPE breaks down. So $\\hat g$ is indeed required for GNPE and we can not simply take $g^\\theta$.
> > >
> > > We are currently working on a revised version of section 3 to make these points clearer, which we aim to post very soon.

---

> > > ### Author Response · Authors · 2021-11-22
> > > **Update of the manuscript**
> > >
> > > Dear Reviewer,
> > >
> > > Thanks for the positive reply that we are on the right track with the past changes. We have now reorganized sections 3.2-3.4 to include a more accessible description of our method, and taking into account clarifying questions received in the review process. We now emphasize earlier that we infer the joint posterior $p(\\theta,\\hat g|x)$ over $\\theta$ and $\\hat g$, and show how this helps us to standardize the pose. We believe that this should clarify your questions. We also added Fig. 2 to illustrate how GNPE alternately samples $g^\\theta$ and $\\hat g$. We further explain more clearly why the approximate pose $\\hat g$ is needed in the first place-- GNPE would not converge without it (see footnote 4).
> > >
> > > Also note the new appendix A.3, in which we show that the posterior estimated with GNPE can indeed be made equivariant by construction.
> > >
> > > Combined with our above reply to your comments, we hope that you are now satisfied with the presentation of our method.  We hope that you can agree that we have “explained all the related conclusions and definitions”, as you had requested  and that this allows you to “consider increasing your score”.

---

> ### Author Response · Authors · 2021-11-21
> **Kind reminder to update your review**
>
> Dear Reviewer, we provided detailed answers to all your comments 1-5 (note in particular the new extensive appendix A.1 and A.2). There have also been several updates in discussion with reviewer tFyT which resulted in them changing their score to 8 ('good paper, accept'), with confidence 4.
>
> We would appreciate a response from you, and in particular whether our response alleviates your concerns and allows you to update your score, or guidance on what additional evidence and clarification you would ask us to provide.

---

> ### Comment · Area_Chair_nuwc · 2021-11-24
> **Respond to author feedback**
>
> Please respond to author feedback and other reviewers' comments and indicate if it changes your rating.

---

### Author Response · Authors · 2021-11-12
**General response to all reviews**

We would like to thank the reviewers for their helpful reviews and generally positive evaluation of our work, pointing out that GNPE is “an interesting method for a methodologically interesting problem” (tFyT), that the paper is “well-written” (tFyT,Xr5B), the “overall execution is of high quality” (Xr5b) and that “GNPE can achieve better performance than the traditional NPE” (W1NJ). We note that all reviewers agree on the technical novelty and convincing empirical results of our method. We emphasize that GNPE leads to a dramatic performance improvement on a challenging ‘real-world’ scientific inference problem of central importance for gravitational-wave astronomy, namely inference of source parameters of binary black hole mergers (component masses, spins, location, and orientation).

The main criticism of the reviewers, and possibly the reason for the restrained scores,  seems to relate to the description of our method. One challenge in presenting this work is that it combines several ideas including simulation-based inference, geometric properties (equivariances), and Gibbs sampling, and further the motivating example involves specialized domain knowledge. To make the paper accessible to a wider audience, we will make sure to improve the presentation and include a compact, high-level motivation and description of our method before the technical details (possibly as a subsection titled “intuition” in 3.3). We will also work to improve the text throughout, particularly at the start of the existing section 3.3 where we introduce the pose proxy $\\hat g$.

Briefly, the central problem that we resolve is **how to simultaneously infer the pose of a signal and use that inferred pose to standardize (or align) the data so as to simplify the analysis**. Pose-standardized data is easier for neural networks to analyze because it is effectively lower dimensional. This is a **circular** problem because one cannot standardize the pose (contained in model parameters $\\theta$) without first inferring the pose from the data; and conversely one cannot easily infer the pose without first simplifying the data by standardizing the pose. Our resolution is to (1) start with a rough estimate of the pose, (2) transform the data based on this, (3) obtain an improved estimate of the pose based on the transformed data, and (4) iterate steps 2 and 3 until convergence.

As noted by Xr5B, the problem we seek to solve is “a completely different creature” from the problems usually solved by G-equivariant normalizing flows. We are interested in **joint transformations of data and parameters**---i.e., **conditional probability distributions** that transform equivariantly under G [equation (4)]. To our knowledge, there is **no existing architecture** that can enforce such transformation properties for normalizing flows.

We hope that these explanations and resulting modifications will make the paper accessible to a broad range of readers, and that this will allow the reviewers to endorse the paper more wholeheartedly.

---

### Author Response · Authors · 2021-11-17
**Updated version of the manuscript**

We posted an updated version of the manuscript, in which we address most of the feedback by the Reviewers. In addition to a few changes to the main part, we added appendices A.1 and A.2 with further derivations for the claims in section 3, and a number of additional plots in appendix C requested by the Reviewers.

We are still working on improving the high-level description of our method in sections 3.3 and 3.4. We are awaiting feedback from the Reviewers whether our explanations posted in the individual comments clarify their questions, and we will post another update based on this discussion.

---

### Author Response · Authors · 2021-11-23
**New update of the manuscript**

We posted another updated version of the manuscript. In particular, we reorganized sections 3.3 and 3.4 in response to the feedback. The description of our method is now easier to follow. We also added in total 3 more appendices (A.1, A.2, A.3) in response to requests for derivations, as well as several supplementary plots.

We believe this process led to a significantly improved presentation of the method, and that this addresses the remaining concerns of the Reviewers. For further details see our latest comments in response to W1NJ and Xr5B. We also highlight the positive discussion with Reviewer tFyT, who came up with a toy example for a further illustration of GNPE.

---

### Decision · Program_Chairs · 2022-01-20

**Decision:**

Accept (Poster)

**Comment:**

This paper studies group equivariant neural posterior estimation which seeks to endow conventional neural posterior estimation method with equivariance of both the data and parameters simultaneously. To test the efficacy of the proposed approach the authors experiment with gravitational wave data and show that the proposed GNPE achieves considerable performance gains.


Strengths:

- The approach is independent of neural architectures and does not necessitate knowledge of exact equivariances.
- The method seems to be much better than regular NPE in cases where there are known equivariances.

Weaknesses:

- the writing of the paper is not clear, which makes the paper difficult to read and evaluate.
- It is hard to check the correctness of the conclusions and algorithm due to lack of necessary assumptions and derivation steps.
- the authors are knowledgeable about the subject but present material in a slightly callous way which prevents a precise understanding of their techniques.


This is a borderline paper with two reviewers in favor of acceptance and two with a slight tendency to reject. The two negative reviewers did not engage in a discussion with the authors or did not complete that discussion, failing to confirm their ratings or provide an update of those ratings. They also do not seem to give strong arguments for rejection. Based on this, I recommend the paper for acceptance. However, I encourage the authors to take into account the reviewers' comments, especially the part on clarity and rigor, to improve the paper for its camera-ready version.